# Ecological properties of soil improved by high–performance ester materials under freeze–thaw cycles conditions

Cuiying Zhou[1,2], Qingxiu Zhang[2], Jin Liao[2], Haoqiang Lai[2], Zhen Liu [2*]

**1** School of Civil and Transportation Engineering, Guangdong University of Technology, Waihuan West Road, Guangzhou University Town, Guangzhou, China, **2** Guangdong Engineering Research Centre for Major Infrastructures Safety, School of Civil Engineering, Sun Yat-sen University, Guangzhou, China

* liuzh8@mail.sysu.edu.cn

## Abstract

High-performance ester materials (HPEMs) are widely applied in slope soil restoration owing to their biodegradability, with a predictable degradation cycle of 2–3 years. Although HPEMs have been extensively studied for tropical applications, their performance in cold regions subject to frequent freeze-thaw cycles (FTCs) remains poorly understood. We hypothesize that FTCs degrade HPEM performance but enhance plant germination. Controlled experiments demonstrated that FTCs reduced material viscosity by 70.5% and water absorption by 52%, while increasing germination rates by 30%, revealing a trade-off between material durability and ecological benefits. Additionally, the field water holding capacity and soil conductivity of the improved soil decrease. Based on these experimental results, the study establishes quantitative relationships between FTCs and soil ecological properties improved by HPEMs. Quantitative relationships revealed linear viscosity decline and exponential water absorption decay trends, with high correlation coefficients ($R^2 \geq 0.95$). This study establishes a comprehensive theoretical framework for predicting the service life of ester materials in cold regions and optimizing their application strategies.

## 1. Introduction

Soil is the cornerstone of terrestrial ecosystems, supporting critical functions such as nutrient cycling, water regulation, and biodiversity maintenance. Consequently, ecological soil restoration has become a paramount global environmental priority [1]. In recent years, polymeric materials have gained prominence in ecological soil remediation strategies, demonstrating significant potential for improving soil structure, optimizing physicochemical characteristics, and enhancing nutrient retention capacity [2–4]. However, field applications face significant challenges, particularly in cold regions where FTCs induce structural degradation and nutrient loss in amended soils [5–8]. Furthermore, the service life of polymer materials is susceptible to FTCs,

**Data availability statement:** All relevant data are within the paper and its Supporting information files.

**Funding:** The research is supported by the National Natural Science Foundation of China (NSFC) (Grant Numbers: 42293354, 42293355, 42293351, 42277131, 42293350). The funders had no role in study design, data collection and analysis, decision to publish, or preparation of the manuscript. There was no additional external funding received for this study.

**Competing interests:** The authors have declared that no competing interests exist.

which affects their effectiveness and efficiency in soil quality improvement. Therefore, systematic investigation of FTCs-induced modifications in soil-amendment polymers provides dual benefits: advancing fundamental understanding of material longevity mechanisms while developing cost-optimized strategies for large-scale ecological restoration initiatives.

Unlike synthetic polymers (e.g., sodium polyacrylate [9–12]) or biopolymers (e.g., cellulose [13–15]), HPEMs exhibit a predictable 2–3 year degradation cycle without generating toxic byproducts [16,17]. However, their long-term performance under FTCs—particularly the interplay between material degradation and ecological benefits—remains unexplored. This study bridges this gap by quantifying how FTCs alter HPEM properties while simultaneously enhancing plant germination, a dual-phase response not previously reported. Prior research has demonstrated that polymeric materials can preserve soil organic matter [18], enhance biomass accumulation [19], and stimulate plant growth and development [20]. These materials exhibit unique soil-amending capabilities, including improved soil mechanical strength and enhanced water retention capacity. These multifunctional attributes have consequently spurred significant scientific interest in their applications for sustainable land management. Ester-based polymers represent a class of soil amendment materials widely utilized in slope stabilization and ecological restoration projects. Ester adhesives predominantly consist of polyvinyl acetate, characterized by long-chain macromolecular structures and abundant hydrophilic functional groups Water-retaining ester formulations primarily comprise sodium polyacrylate, whose cross-linked polymeric networks enable substantial water absorption and storage through hydrogen bonding interactions [21–23]. Ester-based polymers demonstrate significant ecological advantages, including erosion mitigation, soil fertility enhancement [24,25], and increased agricultural productivity [26,27]. These cost-effective materials not only reduce soil remediation expenses but also show broad agricultural applicability due to their multifunctional performance.

However, the performance of polymeric soil amendments is inherently limited by environmental stressors, compromising their soil-stabilization efficacy and accelerating material degradation. This vulnerability has driven substantial research focus on FTCs induced mechanical alterations in polymer-treated soils. Several studies have found that, when subjected to FTCs, soil amended with polymer materials exhibits higher cohesion, internal friction angle, and compressive strength compared to untreated soil [28–30]. Another research study revealed that, after multiple consecutive FTCs, polymer-treated soils exhibited greater frost resistance than untreated soils. This was attributed to the material's hydrophobicity, which mitigated the effects of unfrozen water transport and redistribution in the soil [31–33]. Additionally, some scholars have investigated the durability of polymer materials and amended soils under FTCs. Their findings indicate that polymers remain effective even after numerous successive FTCs [34,35] and can improve soil durability [36]. These studies suggest that polymers can adapt to extreme conditions and are suitable for soil improvement in seasonal permafrost regions. Notably, most existing research on HPEMs has focused on tropical and temperate applications, where freeze-thaw

effects are negligible. The northern expansion of ecological restoration projects using these materials creates an urgent need to understand their FTCs response. Unlike conventional polymers, HPEMs combine biodegradability with unique ecological benefits, making their FTCs durability a critical but unexplored aspect for cold region applications.

Building on prior studies, this work specifically investigates the degradation kinetics of HPEMs under FTCs and their ecological impacts, which remain underexplored. We hypothesize that FTCs will degrade the performance of HPEMs in soil improvement, reducing their water retention and conductivity-enhancing properties, but may simultaneously promote plant germination due to altered soil microenvironments. This study aims to: (1) quantify the degradation of HPEMs under FTCs, (2) evaluate the resulting changes in soil ecological properties, and (3) assess the implications for plant growth. The novelty lies in the systematic decoupling of physical, chemical, and biological effects of FTCs on HPEM-amended soils, providing a framework for optimizing material formulations in cold regions.

This study establishes quantitative relationships (variation laws) between FTCS frequency and the degradation of HPEMs, soil properties, and plant responses, filling a gap in the systematic analysis of HPEMs under cyclic freezing. While germination and height provide initial insights into plant responses, ecological restoration effects were further corroborated by soil physicochemical properties (field water-retention capacity, conductivity, pH) to ensure a holistic assessment of HPEMs' impacts.

## 2. Experimental details

Soil nutrients are fundamental to the ecological properties of soil. The main indicators of soil nutrient storage include the field water-retention capacity, conductivity, and pH of soil. Pedospheric water mediates critical biogeochemical processes, regulating plant-available water content and microbial hydration dynamics central to nutrient cycling [37]. The field water-retention capacity represents the highest soil water content that can be stably maintained and is plant-available water reserves. Soil conductivity, an electrochemical characteristic, can reflect the salinity, nutrient content, moisture content, and other physicochemical properties, serving as an indicator of soil nutrient storage [38]. Changes in soil pH can alter soil nutrients and microbial indicators [39]. Therefore, the impact of FTCs on the water-holding capacity, conductivity, and pH of polymer-amended soils is the systematically evaluates of this research.

In this paper, we conducted indoor experiments including FTCs and plant planting (Fig 1). We investigated the effects of FTCs on the properties of HPEMs and the field water-retention capacity, conductivity, and pH of soil amended soil

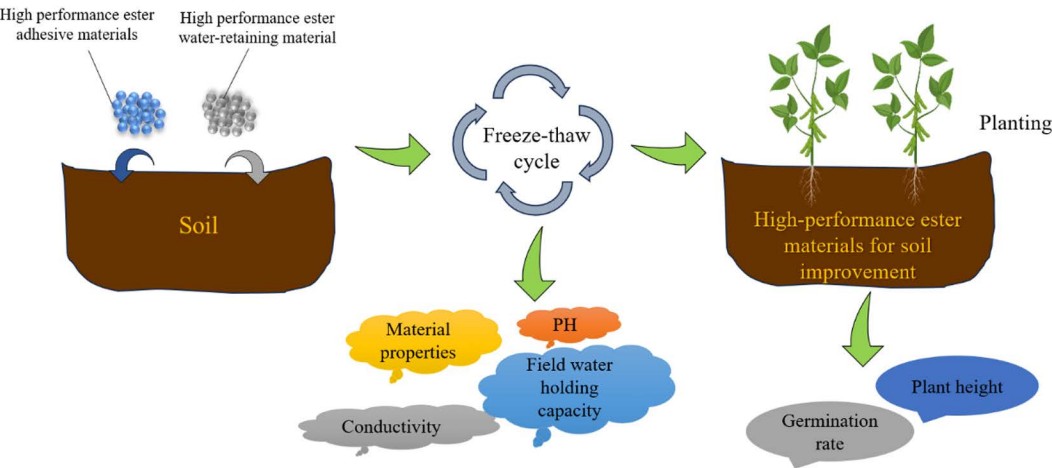

**Fig 1. Schematic illustration of experiment.**

matrices. Subsequently, we examined the effects of FTCs on plant growth. Finally, based on the experimental results, we analyzed the relationship between the number of FTCs and each key response variables.

## 2.1. Experimental materials

The experimental materials comprised soil samples and HPEMs. The soil sample was Quaternary Holocene yellow-brown residual clay, as illustrated in Fig 2a. The detailed specifications of these soil samples are presented in Table 1 [40]. This yellow clay soil was chosen for improvement in the experiment due to its poor scouring resistance.

In this experiment, the HPEMs comprised consisted of high-performance ester adhesive and water-retaining materials. The adhesive materials are water-insoluble modified polyesters. At room temperature, they appear as a white, viscous emulsion, exhibiting dispersibility in water. When mixed with water in specific proportions, they can be utilized as a dispersion, as depicted in Fig 2b. The water-retaining materials, made from HPEMs, are resin-based and exhibit high water-absorption capabilities. Their main constituent is sodium polyacrylate, which consists of fine, white particles dried at room temperature, as shown in Fig 2c. Upon absorbing water, these particles transform into a transparent gel, with a water absorption multiplication rate of up to 250%. Notably, the materials employed in this experiment are biodegradable under natural conditions. Specifically, they have a degradation cycle of two to three years, produce $CO_2$ and $H_2O$ as degradation products, and do not generate toxic or hazardous substances [16,17].

The selection of pigeon pea (Cajanus cajan) was based on its well-documented performance in our prior studies [21–23,41,42]. This species demonstrates strong adaptability to diverse environments, including drought-prone and

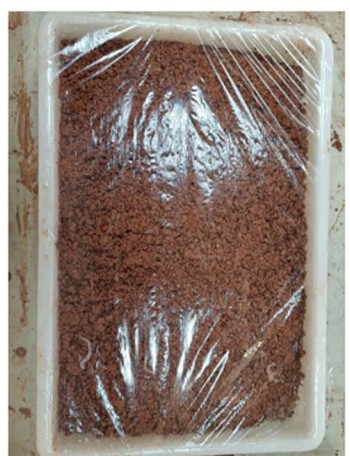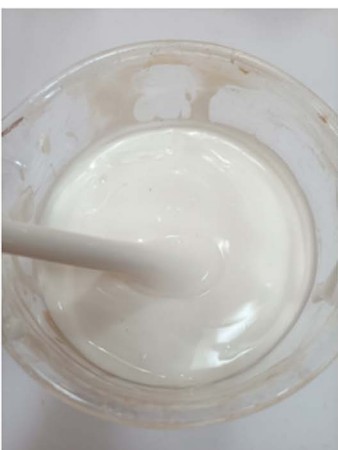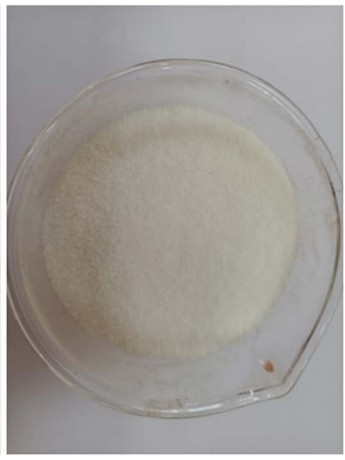

(a) Experimental soil sample

(b) High-performance ester adhesive materials

(c) High-performance ester water-retaining materials

**Fig 2. Experimental materials.**

**Table 1. The specifications of the experimental soil samples.**

| Soil type | Sand content (%) | Natural water content (%) | Air–dried water content (%) | Natural density (g/cm³) | Dry density (g/cm³) | liquid limit (%) | plastic limit (%) |
|---|---|---|---|---|---|---|---|
| Yellow clay soil | 0.45±0.15 | 25.26 | 3.59 | 1.52 | 1.21 | 54.1 | 36.1 |

(Notes: Sand content measured by sieving (ISO 11277); liquid/plastic limits by Casagrande method (ASTM D4318).)

nutrient-poor soils, and its deep root system enhances soil stabilization and erosion control. Additionally, pigeon pea is cost-effective to cultivate, aligning with the goal of reducing restoration costs. Based on these favorable attributes, pigeon pea was chosen as the plant species for this experiment.

The authors declare that the study on plants in this research, including the collec-tion of plant materials, complies with relevant institutional, national, and interna-tional guidelines and legislation.

## 2.2. Experimental methods

First, soil samples were collected (the sampling point: 23°4′16.3″N, 113°23′29.9″E). Undisturbed soil samples (0–30 cm depth) were obtained via stratified sampling using stainless steel shovels. Stratified sampling was performed by digging a profile with a shovel. For multi-point mixed samples, it can be repeatedly discarded by the quartering method to ensure the representativeness of the sample.

This research aimed to investigate the impact of freeze–thaw cycles on the properties of HPEMs and the ecological properties of soils amended with these substances. To achieve this, we conducted freeze–thaw cycles experiments. Additionally, a planting experiment was carried out to assess the effects of soil amended with high-performance ester substances on plant growth after undergoing multiple FTCs.

**2.2.1. FTCs experiment.** The FTCs experiments were designed hierarchically: Category I quantified the intrinsic degradation of HPEMs (materials alone); Category II evaluated soil properties amended with individual HPEM components; Category III tested composite HPEM formulations to simulate field conditions. This approach ensured progressive validation from material to ecosystem levels.

Freeze-thaw cycles were conducted in a climate chamber (ESPEC PSL-3K, ± 0.5°C accuracy). Each FTCs consisted of 12 hours of freezing at −20°C followed by 12 hours of thawing at 25°C, with a total of 15 cycles. Soil samples were mixed with HPEMs at ratios listed in Tables 2 and 3. Viscosity was measured using an NDJ-8S viscometer, and water absorption was determined via the tea bag method [43,44]. Soil conductivity and pH were measured after each FTCs.

Category I: "Material Property Degradation Under FTCs". Key physicochemical properties of adhesive materials encompass viscosity and electrical conductivity characteristics. The properties of adhesive materials, specifically their viscosity, were measured using an NDJ–8S rotational viscometer (Fig 3a). The water-retention capacity of water-retaining materials was evaluated by the tea bag method [43,44], which determines their water-absorption multiplicity, a crucial indi-cator for assessing water retention. For the experiment, specific amounts of high-performance ester adhesive material and water-retaining material were weighed. Their respective viscosities and water-absorption multiplicities were then estab-lished. The samples were sealed and subjected to freezing in a constant temperature and humidity test chamber (Fig 3). The freezing temperature was set at –20°C, based on the average winter temperature in northern China, for a duration of 12 hours [45,46]. Thawing was conducted at 25°C for another 12 hours to ensure complete phase transition. Fifteen FTCs were performed to simulate the material's exposure to multiple annual freeze-thaw events over its 2–3 year degradation

Table 2. Additions of high–performance ester adhesive materials and water–retaining materials.

| Group number | Soil (g) | Adhesive materials (g/m²) | Water–retaining materials (g/m²) |
|---|---|---|---|
| Control group | 500 | 0 | 0 |
| A–1 | 500 | 10 | 0 |
| A–2 | 500 | 20 | 0 |
| A–3 | 500 | 30 | 0 |
| B–1 | 500 | 0 | 20 |
| B–2 | 500 | 0 | 40 |
| B–3 | 500 | 0 | 60 |

**Table 3. Additions to high–performance ester composite materials.**

| Group number | Soil (g) | High–performance ester adhesive materials (g/m²) | High–performance ester water–retaining materials (g/m²) |
|---|---|---|---|
| Control group | 500 | 0 | 0 |
| C–1 | | 10 | 20 |
| C–2 | | | 40 |
| C–3 | | | 60 |
| C–4 | | 20 | 20 |
| C–5 | | | 40 |
| C–6 | | | 60 |
| C–7 | | 30 | 20 |
| C–8 | | | 40 |
| C–9 | | | 60 |

cycle, as observed in seasonal permafrost regions. Considering the material degradation cycle and the average annual freeze-thaw occurrences in the northern region, the experiment was designed to include 15 FTCs, as illustrated in Fig 4. After each cycle, the viscosity of the adhesive materials and the water-absorption multiplicity of the water-retaining materials were re-measured. To ensure accuracy and minimize errors, each treatment in the experiment included three parallel groups. Data are expressed as mean ± standard deviation of three replicate experiments. One-way analysis of variance (ANOVA) was used to test the significance of differences between groups.

Category II: "Soil Property Changes with Individual HPEM Components". This study evaluated FTCs effects on the ecological characteristics of yellow clay soil amended with high-performance ester adhesives or water-retention additives. Seven experimental groups were designed: a control group containing unmodified yellow clay soil, and six treatment groups with soil amendments. The remaining six groups were received with specific weights of high–performance ester adhesives and water–retaining materials, as detailed in Table 2. The following ratios are mainly derived from previous research experience and the engineering practice of this research group [21–23].

Experimental soil samples were prepared through mechanical crushing followed by sieving using a 5-mm mesh sieve. Processed samples were weighed for mass uniformity prior to sequential placement into seven standardized soil-retention containers. Amendment quantities of adhesives and water-retention agents were calculated according to the specifications in Table 2. A dispersion liquid was formed by stirring the adhesive material and water evenly in a 1:20 ratio, which was subsequently sprayed onto the soil samples. Water-retaining materials (sodium polyacrylate, particle size 0.5–1 mm) were homogenized with soil using a mechanical mixer at 200 rpm for 10 minutes to ensure uniform distribution. All treatment groups maintained their native soil moisture content throughout the experimental procedures. The samples were then sealed and subjected to a freeze–thaw cycles process in a constant temperature and humidity test chamber, following the same procedure as in the Category I experiment. This process involved a total of 15 FTCs. After each cycle, the field water-retention capacity, conductivity, and pH of each treatment group were measured. To minimize errors, three parallel groups were established for each treatment (the results are based on the averages of three parallel groups).

Category III: "Composite HPEM Effects on Soil and Plant Systems". We conducted an investigation into the effects of FTCs on the ecological properties of soils improved with various proportions of high-performance ester composites. The experimental design included ten treatments: an unamended control group native yellow clay soil and nine experimental groups incorporating HPEC amendments at ratios specified in Tables 3 and 4. The following ratios are mainly derived from previous research experience and the engineering practice of this research group [21–23].

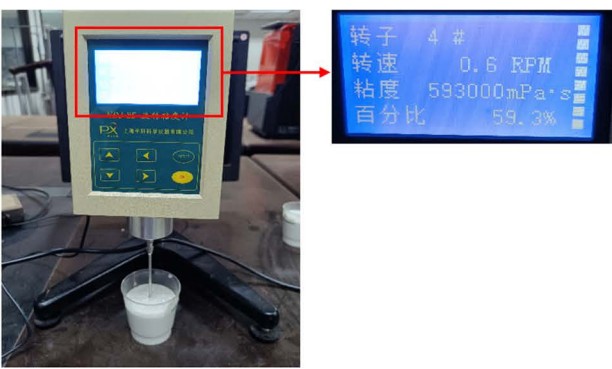

(a) Viscometer

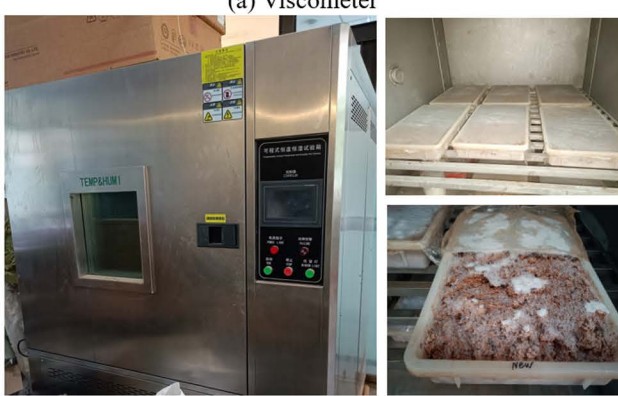

(b) Chambers with constant temperature, humidity, and frozen soil samples

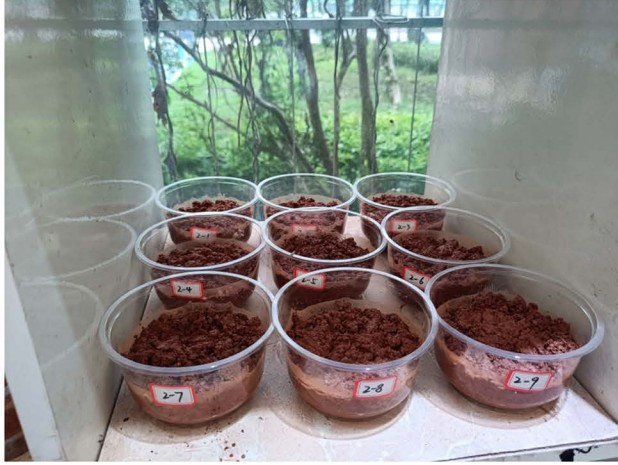

(c) Soil samples

**Fig 3. Experimental equipment and specimens.**

Soil specimens were processed using identical protocols to those implemented in the Category II experiments. Subsequently, they were weighed to an equal mass and placed sequentially in ten soil boxes. The specific quantities of adhesive and water-retaining materials were calculated according to the material ratios specified in Table 2. To prepare the adhesive solution, the adhesive material was mixed with water in a 1:20 ratio. The water-retaining materials were evenly sprinkled

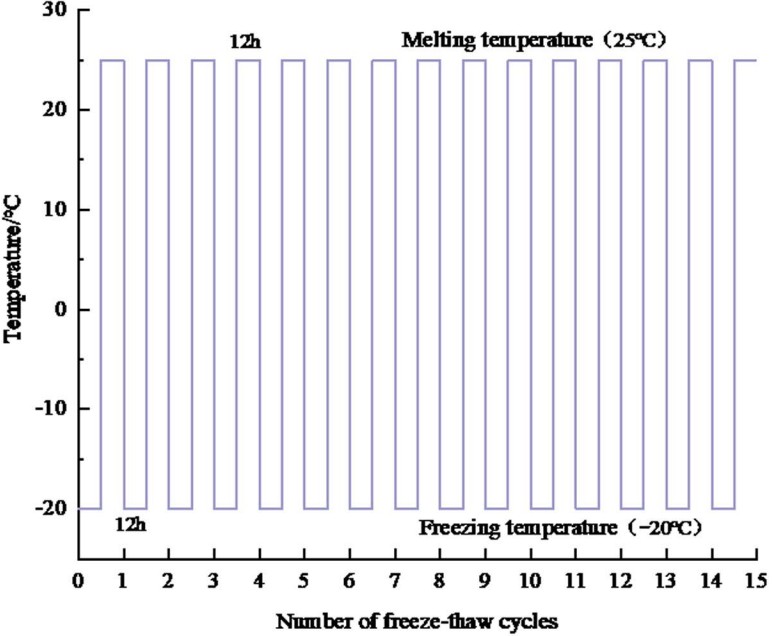

**Fig 4. Schematic diagram of 15 FTCs process.**

onto the soil samples and thoroughly mixed. The adhesive solution was then uniformly dispersed into the soil samples. The water content of the soil samples in all treatment groups was adjusted to the natural level. The freeze–thaw cycles process, number of cycles, and soil sample indices were determined in the same manner as in the Category II experiment.

**2.2.2. Planting experiment.** Planting experiments were conducted to investigate the effect of improved soil on plant growth after the FTCs. The experiment was performed under controlled conditions to ensure consistency: light exposure was maintained at 12-hour light/dark cycles using artificial grow lights (intensity: 200 µmol/m²/s), and temperature was regulated at 25°C during the day and 20°C at night [47,48]. Nutrient control was achieved by applying a standardized nutrient solution (Hoagland's solution) every 3 days to avoid nutrient variability. Planting depth was uniformly set at 2 cm for all seeds to ensure consistent germination conditions.

The estimation of germination rate followed these steps:

(1) Seed pretreatment: Healthy, uniformly sized seeds of pigeon pea were selected and soaked in distilled water for 12 hours to promote water absorption and expansion.

(2) Planting and observation: The pretreated seeds were planted evenly in soil containing different amendment materials. Temperature and humidity were monitored daily to maintain optimal conditions (70% relative humidity). Germination time and the number of germinated seeds were recorded.

(3) Calculation of germination rate: On the 7th day after planting, the number of germinated seeds in each treatment group was counted. Three parallel groups were established for each treatment to minimize errors. The germination rate was calculated as the average of these parallel groups.

(a) Viscometer

(b) Chambers with constant temperature, humidity, and frozen soil samples

(c) Soil samples

**Table 4. Comparison of lab-based FTCs conditions with real-world data from northern China.**

| Parameter | Lab Simulation | Northern China (Annual Average) | Reference |
|---|---|---|---|
| Freezing Temperature | −20°C | −15 to −25°C | [45] |
| Thawing Temperature | 25°C | 20–30°C | [46] |
| FTCs Frequency | 15 cycles | 10–20 cycles | Regional climate reports |

**2.2.3. Conductivity, pH and water retention measurements.** Soil conductivity measurements were performed in strict accordance with the *Standard Method for the Determination of Soil Conductivity* jointly promulgated by the International Soil Science Society (ISSS) and the Soil Science Society of America (SSSA). This standard provides comprehensive protocols for soil specimen preparation, electrode selection/calibration, measurement procedures, and data analysis. For the determination of soil pH, we followed the Method for the Determination of Soil pH issued by the International Organization for Standardization (ISO) [49], which covers the collection and preparation of soil samples, the preparation of measurement solutions, the calibration and use of pH meters, and the reporting of measurement results. As for the determination of soil water retention, we have adopted the "Method for Determining the Characteristic Curve of Soil Moisture" recommended by the Food and Agriculture Organization of the United Nations (FAO) [50], which comprehensively assesses the water retention performance of soil by determining the water content of the soil under different suctions. Rigorous adherence to these internationally recognized standards ensured metrological traceability of conductivity, pH, and water retention measurements, thereby establishing a robust technical foundation for data interpretation and applied research.

**2.2.4. Statistical analysis and data quality control.** Data were analyzed by one-way ANOVA with Tukey's post-hoc test (SPSS 26.0). Significance: $p < 0.05$. Triple parallel samples were used; instruments calibrated daily.

## 3. Experimental results and analysis

### 3.1. Effects of FTCs on HPEMs and amended soils

**3.1.1. Effects of FTCs on material properties.** Before FTCs, the high-performance ester adhesive materials exhibited initial viscosity and conductivity values of 525 Pa·s and 1743 µs/cm, respectively. Fig 5 illustrates the trend in viscosity as the number of FTCs increased. The viscosity exhibited a linear decline during the first 10 cycles (slope = −36.7 Pa·s/cycle, $R^2 = 0.98$), followed by a plateau phase (Fig 5), indicating structural saturation. Notably, the rate of viscosity reduction decelerated after the 11th cycle, indicating structural damage saturation. In contrast, the conductivity of the adhesive materials fluctuated minimally throughout the FTCs and remained almost unchanged at 1732 µs/cm after 15 cycles, indicating minimal impact on electrical conductivity.

The high-performance ester-based water-retaining polymer exhibited an initial water-absorption capacity of 250 g/g prior to FTCs exposure. As shown in Fig 5b, progressive FTCs treatment resulted in a systematic decrease in the water-absorption capacity. Specifically, the first FTCs induced the most pronounced attenuation, with the absorption capacity decreasing by up to 9.2%. Water-absorption capacity decreased exponentially ($y = 250e^{-0.06x}$, $R^2 = 0.95$), reaching 120 g/g after 15 cycles, likely due to polymer chain scission (Fig 5). After the 10th FTCs, the material exhibited a progressive decline in water absorption capacity, ultimately reaching a stabilization plateau. These findings indicate that FTCs significantly reduce the water-absorption capacity of high-performance ester water-retaining materials.

Changes in the properties of high-performance ester adhesives and water-holding materials after FTCs can be attributed to: Repeated FTCs induce microcracks in the polymer matrix, leading to irreversible structural damage. The observed exponential decay in water absorption capacity ($y = 250e^{-0.06x}$) quantitatively demonstrates polymer chain

scission processes. Specifically, FTCs-driven degradation disrupts the cross-linked sodium polyacrylate network through hydrolytic cleavage of ester bonds, thereby diminishing its water-retention capability.

### 3.1.2. Effects of FTCs on field water–retention capacity of improved soils.
The field water-retention capacity, which is the maximum soil water content conducive to effective plant growth, was investigated in relation to the impact of FTCs on improved soils. The results are presented in Fig 6.

As the number of FTCs increased, the field water-retention capacity of soil samples in each treatment group gradually decreased. Despite a 12.2% reduction after 15 cycles, ester-amended soils retained 39.5% water content—significantly higher ($\rho < 0.01$) than the control group's 28.3% (Fig 6).

In soil treatments using high-performance ester adhesives or water-retaining agents alone, the B-3 group demonstrated superior water-retention capacity: 45.8% under FTCs-free conditions and 39.5% following 15 FTCs. The application of high-performance ester adhesives or water-retaining materials significantly increased the soil's field water-retention capacity.

Comparative analysis of ester composite formulations revealed that the C-9 formulation maintained peak water retention 39.5% post-15 FTCs, whereas C-1 showed no significant improvement over baseline control: 22.3%. The data demonstrate that FTCs exposure degrades water-retention performance in modified soils, reducing the efficacy enhancement from ester composites by 18–27% relative to FTCs-free conditions. However, the improvement effect of the materials remained significant even after 15 FTCs.

The observed decrease in field water-holding capacity of amended soils can be attributed to: FTCs induces water migration within soil matrices, triggering structural reorganization that compromises moisture retention capabilities.

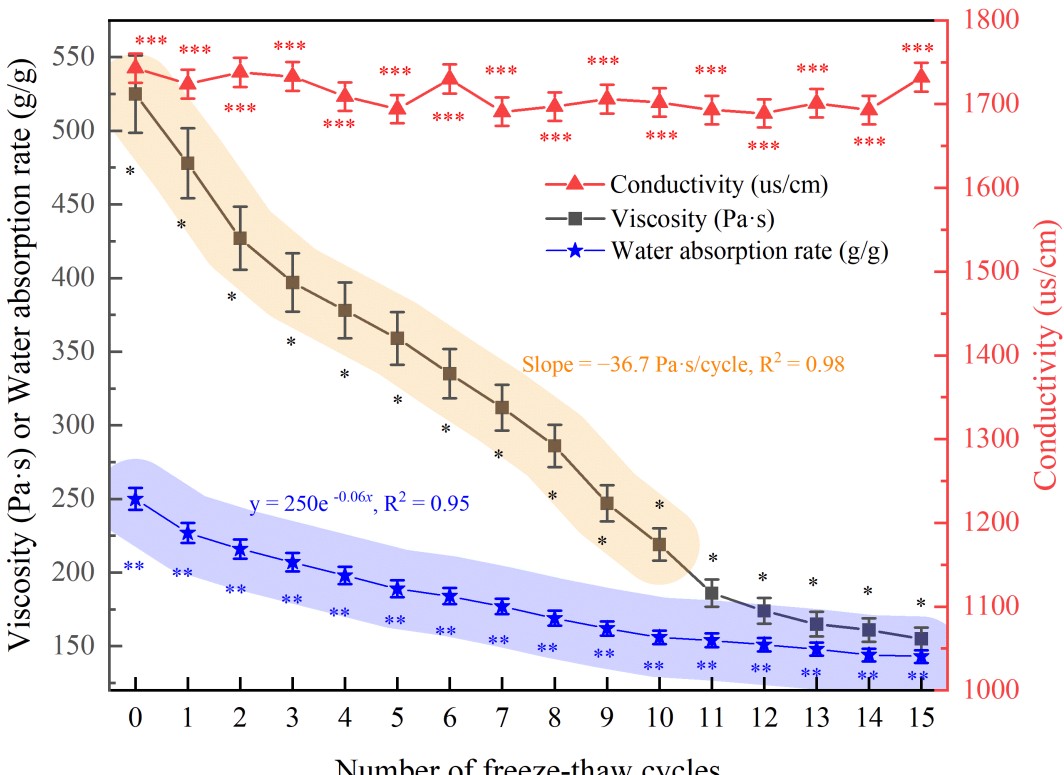

**Fig 5. Changes in properties of HPEMs under different numbers of FTCs.** (Note: Data are expressed as mean±standard deviation (n=3)).

### 3.1.3. Effects of FTCs on conductivity of improved soils.

Soil conductivity, a crucial indicator of soil nutrient storage and the presence of inorganic salt nutrients, undergoes variations due to FTCs. Fig 7 presents the conductivity changes in the improved soil subjected to these cycles. Across all treatment groups, conductivity decreased as the number of FTCs increased.

The B-3 treatment group demonstrated the highest electrical conductivity when either high-performance ester-based adhesives or water-retaining agents were individually applied to soil. Conductivity values measured 516 µS/cm prior to freeze–thaw cycles and 415 µS/cm after 15 FTCs. These values were 73.2% and 53.7% higher than the control group's conductivity. Conductivity reduction (19.6%) correlated strongly with adhesive material degradation (Pearson's r = 0.89, p < 0.01), suggesting ion loss via polymer breakdown.

As demonstrated in Fig 7b, varying proportions of HPEM exhibited differential impacts on soil electrical conductivity. The electrical conductivity measurements across all experimental groups consistently exceeded those recorded in the control group. Maximum conductivity values were observed in the C-3, C-6, and C-9 treatment groups, each amended

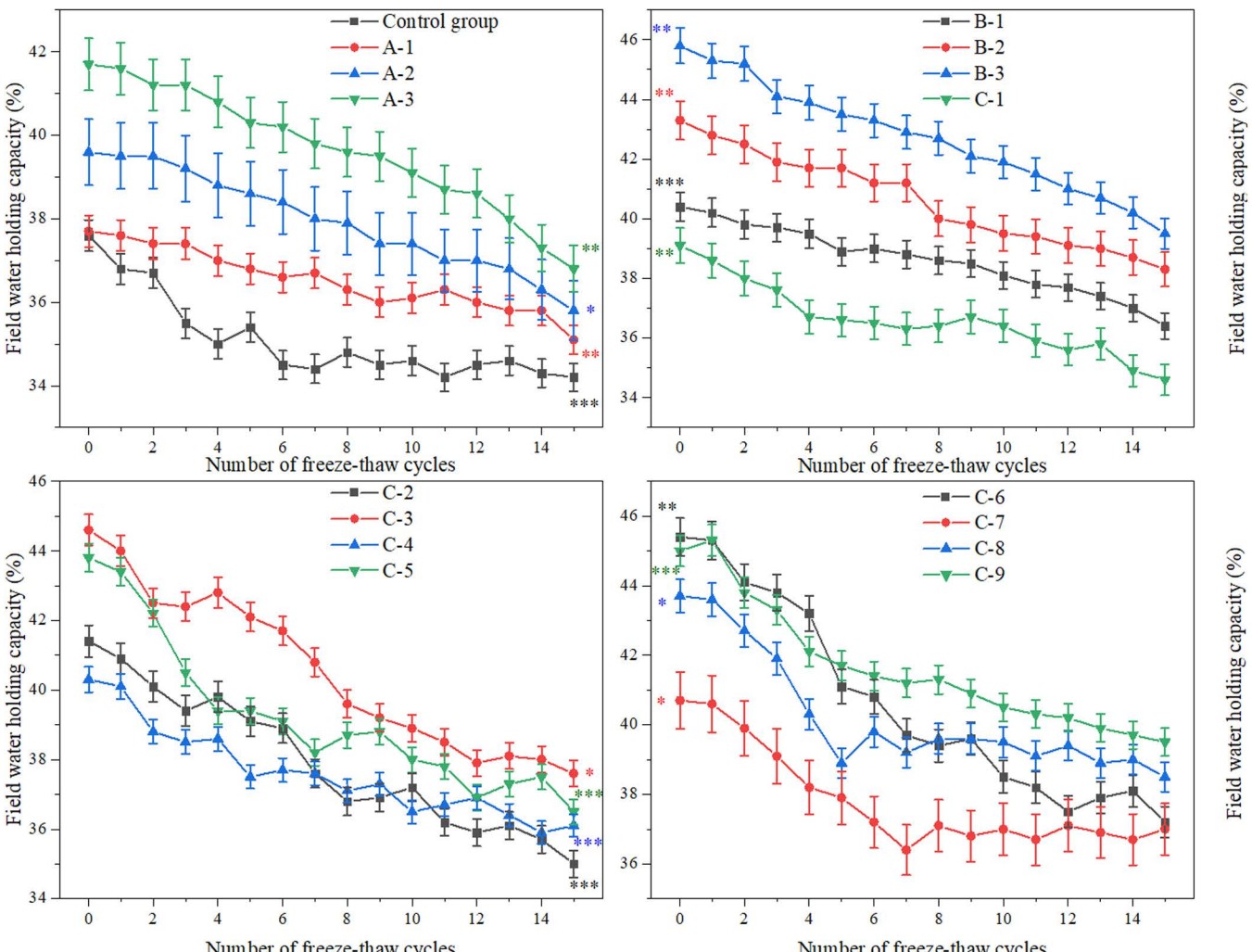

**Fig 6. Changes in field water–retention capacity of improved soils under different numbers of FTCs.** (Note: Data are expressed as mean ± standard deviation (n = 3); Groups: A (adhesive), B (water-retaining agent), C (composite).).

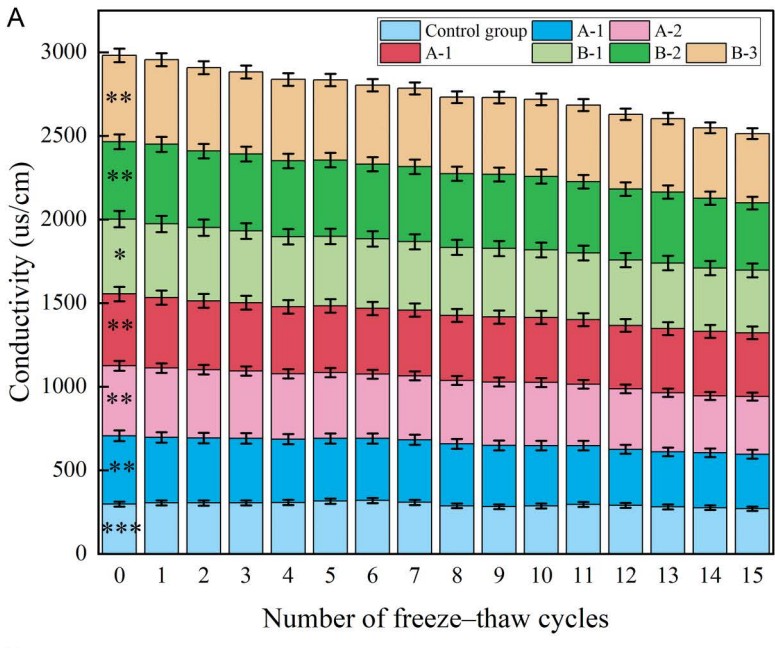

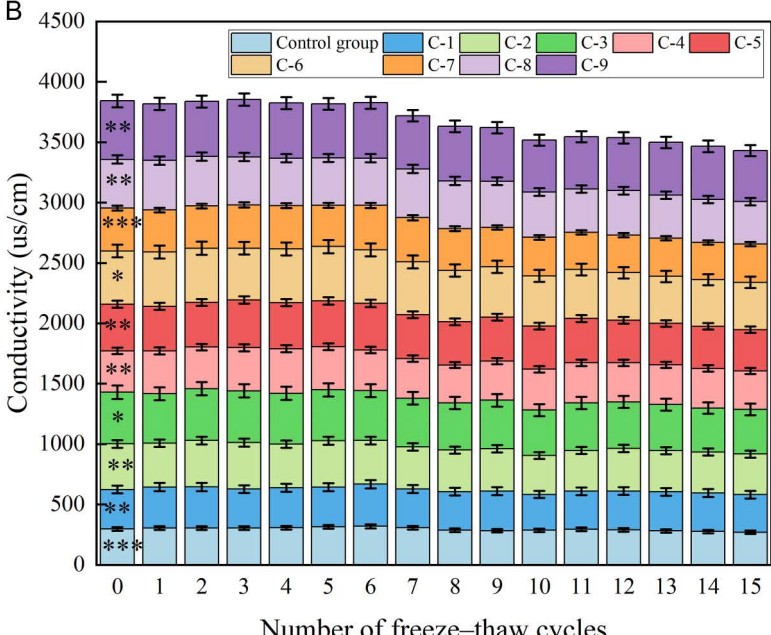

**Fig 7. Changes in conductivity of improved soil under different numbers of FTCs.** (Note: Data are expressed as mean ± standard deviation (n = 3); The figure plotted is a stacked plot, with the value of each data being the upper value minus the lower value; Groups: A (adhesive), B (water-retaining agent), C (composite); *p < 0.05 vs. control group).

with 60 g/m² of high-performance ester-based water retention agents. These findings indicate a substantial conductivity enhancement mediated by the water retention additives.

Following 15 FTCs, the C-9 group retained the highest conductivity (423μs/cm), representing a 12.8% reduction from pre-cycling levels. This demonstrates the capacity of HPEMs to enhance soil conductivity, albeit with a gradual reduction

in efficacy proportional to FTCs frequency. Notably, the materials retained functional integrity through 15 cycles, evidencing remarkable durability under repeated freeze-thaw stress. The observed conductivity enhancement showed positive correlation with application rates of the additives.

(a)High–performance ester adhesives and water–retaining materials

(b)High–performance ester composites of different ratios

The reduction in the conductivity of amended soils can be attributed to the following mechanisms: FTCs could enhance ion mobility within the soil matrix, consequently modifying its electrical conductivity.

### 3.1.4. Effects of FTCs on pH of improved soils.

Soil pH is a critical indicator of nutrient availability and microbial activity. Fig 8 presents the pH variations across different treatment groups exposed to FTCs. Notably, the control group exhibited a peak pH after each FTCs. Conversely, the pH of the other treatment groups decreased gradually (indicating mild acidification) with increasing material addition, stabilizing within a near-neutral range of 6.6 to 7.4, likely due to the proton release from hydrophilic groups in HPEMs. Across increasing FTCs, the control group's pH remained stable, whereas the pH of the treated groups increased gradually.

When only high–performance ester adhesive materials or water–retaining materials were applied to the soil, the pH of all treatment groups increased significantly after the ninth FTCs, as depicted in Fig 8a. Subsequent to the 10th cycle, gradual pH attenuation occurred, though values persisted above baseline (pre-FTCs) levels. Fig 8b demonstrates that, under different ratios of HPEMs, the pH of each treatment group rose substantially after the 6th and 11th FTCs. Post 15 cycles, soil pH in treated groups stabilized at 7.1±0.2, aligning with the control group's 7.0±0.1, confirming the transient nature of HPEMs' pH buffering effect.

These findings indicate that the materials lowered the soil pH, and the FTCs reduced their effectiveness. Consequently, after 15 FTCs, the ameliorating effect of the materials gradually diminished, ultimately resulting in pH reversion.

(a)High–performance ester adhesives or water–retaining materials

(b)High–performance ester composites of different ratios

## 3.2. Effects of FTCs on plant growth

Plant growth was determined based on a plant growth experiment, as shown in Fig 9.

### 3.2.1. Effects of FTCs on plant germination rates.

Fig 10 demonstrates the impact of soil amendments on seed germination under varying FTCs frequencies. Incorporation of HPEMs enhanced germination rates across all experimental groups compared to unamended controls. Germination patterns exhibited cyclical fluctuations corresponding with FTCs progression in all treatments. During initial FTCs exposure (cycles 1–5), all amended groups showed reduced germination rates relative to non-frozen controls. Subsequent to the 5th FTCs, gradual germination recovery commenced in all treatment groups. Fig 10b reveals pronounced germination enhancement in C-1 through C-9 following the 11th FTCs, with rates significantly exceeding control values. Throughout the experimental sequence, amended soils consistently maintained superior germination performance relative to controls. The most marked improvement emerged post-15 FTCs, where amended soils demonstrated 23.6% higher germination rates than controls, confirming HPEMs' cryoprotective efficacy on seed viability.

(a)High–performance ester adhesives or water–retaining materials

(b)High–performance ester composites of different ratios

### 3.2.2. Effects of FTCs on plant height.

Fig 11 demonstrates the effects of soil amendments with different materials on plant height under varying FTCs frequencies. The results reveal that, following the introduction of these materials, there was

a modest increase in plant height across all treatment groups. However, this increase was not statistically significant when compared to the control group. Furthermore, after undergoing 15 FTCs, the plant height in each treatment group remained largely unchanged compared to its pre-cycle state. These findings suggest that the addition of HPEMs does not significantly alter plant height, and the presence of FTCs has no discernible impact on plant growth in terms of height.

(a)High–performance ester adhesives or water–retaining materials

(b)High–performance ester composites of different ratios

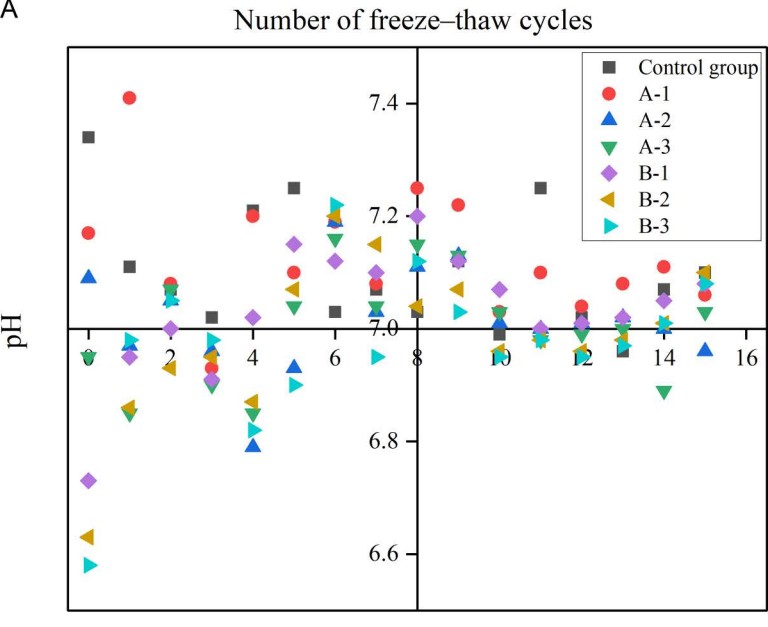

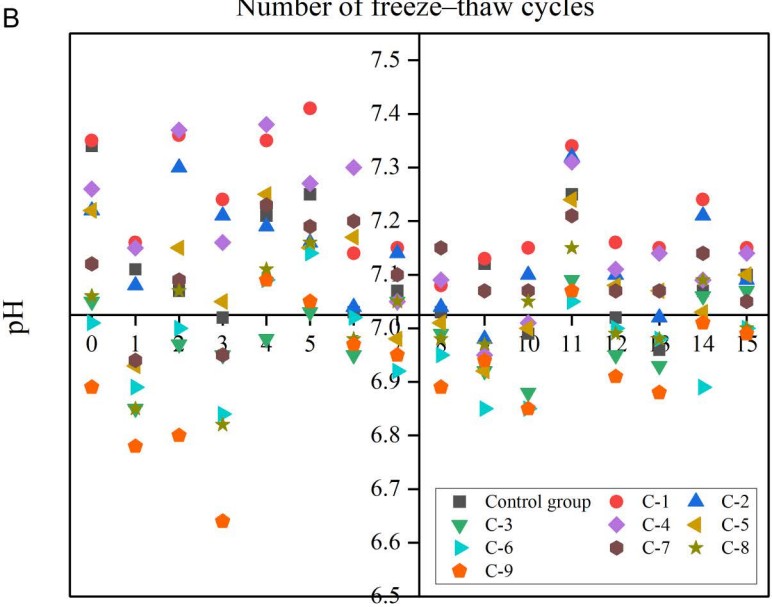

**Fig 8. Changes in pH of improved soil under different numbers of FTCs.** (Note: Decreased pH values reflect mild acidification toward neutrality; Groups: A (adhesive), B (water-retaining agent), C (composite)).

Although the plant height of each treatment group increased slightly with the addition of material, the increase was not significant compared to the control group, and there was no significant difference in the change in plant height between the treatment groups after several FTCs. This phenomenon could be attributed to two principal factors:

Material effect: the change in soil environment by the HPEMs may not be sufficient to significantly affect plant growth.

Freeze-thaw effect: The effect of FTCs on plant growth may be more in the early stage of growth, and as the plant grows, its adaptability to the environment gradually increases, so the height change in the later stage is not significant.

## 4. Discussions

### 4.1. Relationship between number of FTCs and each of HPEMs, soil, and plant growth

The interplay between FTCs-induced material degradation and ecological outcomes was evaluated through both plant responses (germination, height) and soil properties (water retention, conductivity, pH), the latter serving as proxies for nutrient availability and microbial habitat suitability.

**4.1.1. Relationship between the number of FTCs and field water–retention capacity.** Fig 12 illustrates the progressive effects of increasing FTCs frequency on water retention capacities, comparing both the materials' performance and the modified soil's field capacity. With an increase in the number of FTCs, both the water-retention capacity of the materials and the soil field water-retention capacity exhibited a decrease. Additionally, by analyzing S1 Fig in S1 File, a functional relationship can be established between the water-retention capacity of HPEMs and the field water-retention capacity of the improved soil.

$$y_{1\ improved\ soil} = 0.6m_1 y_{1\ materials} + y_{1\ original\ soil} \qquad (1)$$

In the formula above,

$m_1$ is the amount of material in kg/m²;

$y_{1\ materials}$ is a function of the water–retention capacity of materials and the number of FTCs, i.e., $y_{1\ materials} = -6.5926x + 229.13$, where $x$ is the number of FTCs;

$y_{1\ original\ soil}$ is a function of the number of FTCs and the field water–holding capacity of soil with no added material, i.e., $y_{1\ original\ soil} = -0.1835x + 36.476$, where $x$ is the number of FTCs;

$y_{1\ improved\ soil}$ is a function of the amount of water retained in the field of improved soils and the number of FTCs, which can be expressed as $y_{1\ improved\ soil} = -0.1835(21.556m_1 + 1)x + 137.478m_1 + 36.476$, where $x$ is the number of FTCs.

The empirical formula quantitatively expresses the relationship between the amended soil's field water-retention capacity and controlling parameters. This relationship is influenced by the amount of HPEMs and the number of FTCs, with high correlation coefficients supporting its validity. Notably, the field water-retention capacity of the improved soil is most

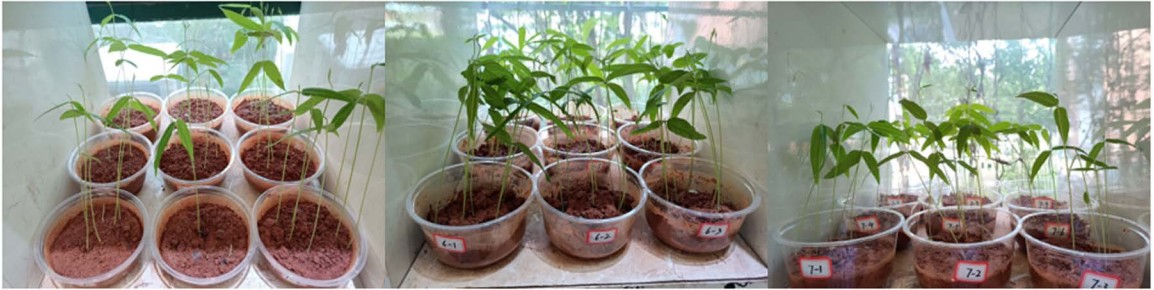

**Fig 9. Plant growth.**

sensitive to variations in the material amount. As the material amount increases, the field water-retention capacity also increases. This aligns with studies showing that FTCs disrupt soil pore structure and polymer-water interactions [51], while increased material dosage enhances water retention through hydrogel network formation [52].

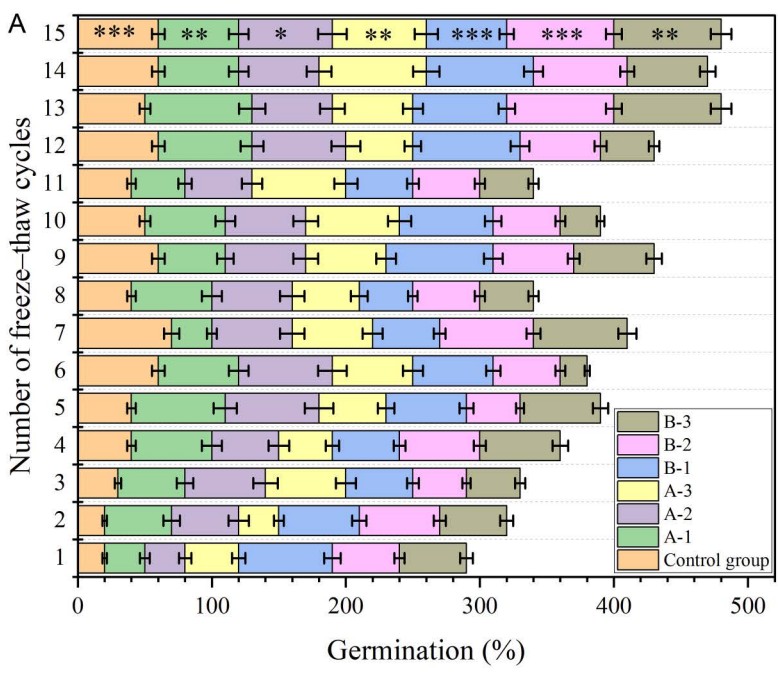

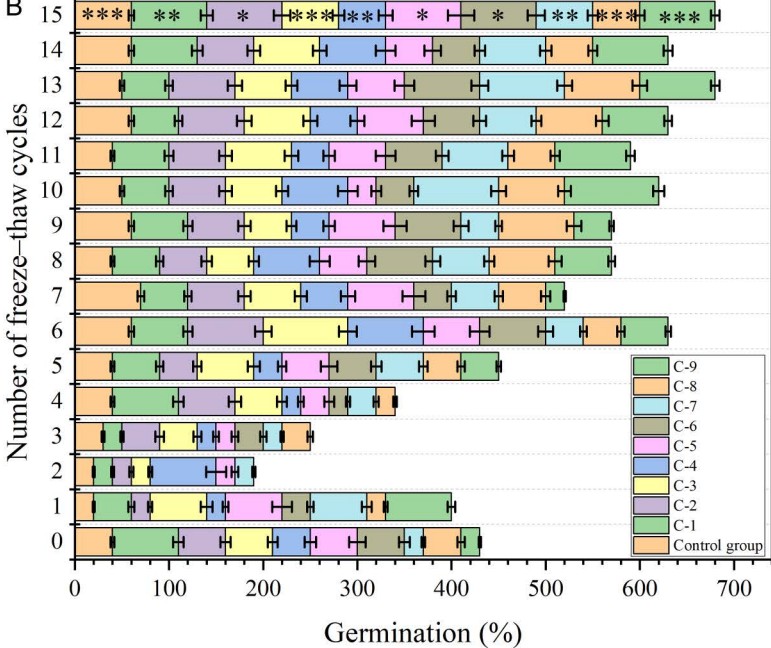

**Fig 10. Changes in plant germination rate under different numbers of FTCs.** (Note: Data are expressed as mean ± standard deviation (n = 3); The figure plotted is a stacked plot, with the value of each data being the upper value minus the lower value.).

**4.1.2. Relationship between number of FTCs and conductivity.** The conductivity of materials and soil conductivity, as shown in S2 Fig in S1 File, decreases as the number of FTCs increases. The functional relationship between the conductivity of HPEMs and the conductivity of the improved soil is depicted in Fig 11.

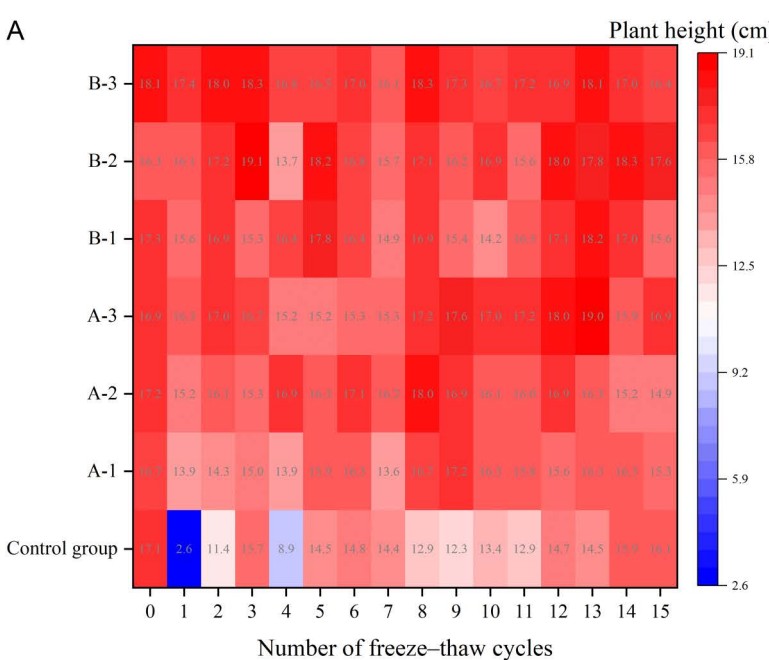

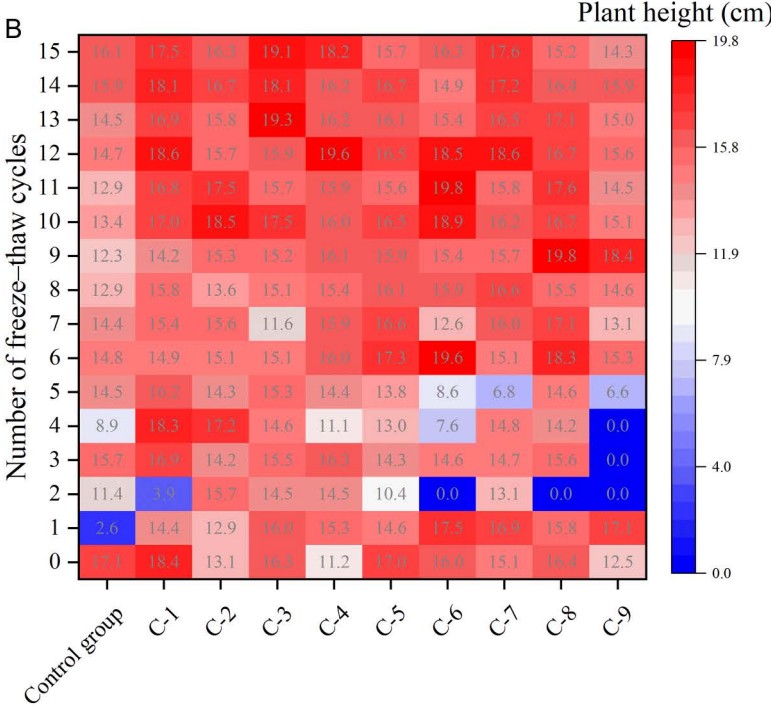

**Fig 11. Changes in plant height under different numbers of FTCs.** (Note: Data are expressed as mean±standard deviation (n=3)).

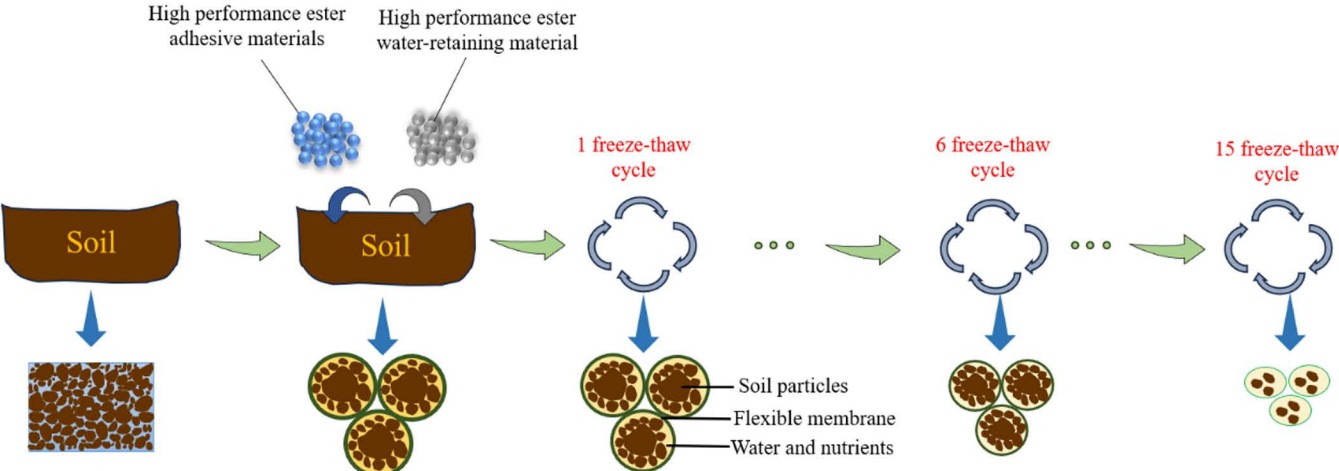

**Fig 12. Mechanism related to effect of FTCs on soil improved by HPEMs.** (chematic based on SEM-observed aggregate disintegration and FTIR-confirmed bond cleavage in HPEMs from [21–23,59,60]).

$$y_{2\ improved\ soil} = 28m_2 y_{2\ materials} + y_{2\ original\ soil} - 1727.58 \times 26m_2 \qquad (2)$$

In the formula above,

$m_2$ is the amount of material in kg/m$^2$;

$y_{2\ materials}$ is a function of the conductivity of materials and the number of FTCs, i.e., $y_{2\ materials} = -2.2191x + 1727.58$, where $x$ is the number of freeze–thaw;

$y_{2\ original\ soil}$ is a function of the number of freeze–thaw for the conductivity of the soil with no added material, i.e., $y_{2\ original\ soil} = -2.2735x + 312.551$, where $x$ is the number of FTCs;

$y_{2\ improved\ soil}$ is the functional relationship between the conductivity of the improved soil and the number of FTCs, which can be expressed as $y_{2\ improved\ soil} = -2.2735\,(27.33m_2 + 1)x + 3455.16m_2 + 312.551$, where $x$ is the number of FTCs.

The conductivity of the improved soil varies with both the number of FTCs and the quantity of materials added. Notably, the conductivity of the improved soil is most responsive to changes in material quantity. Specifically, as the amount of material increases, the conductivity of the improved soil also increases. Such ion loss correlates with FTC-induced polymer degradation, consistent with observations in polyacrylate-treated soils [53]. The positive conductivity-material relationship mirrors findings on ionic polymer amendments [54].

These equations represent empirical trends linking FTCs to soil performance, supported by high correlation coefficients.

**4.1.3. Relationship between number of FTCs and plant germination rate.** S3 Fig in S1 File presents the impact of FTCs frequency on seed germination rates. Germination rates exhibited progressive enhancement with increasing FTCs frequency, though HPEMs efficacy diminished after 5 cycles, inducing treatment group convergence beyond this threshold. Experimental data indicated that germination rates during the first five FTCs were lower compared to those observed in subsequent cycles.

Excessive water retention (>40%) and high conductivity (>500 µS/cm500µS/cm) create hypoxic conditions and osmotic stress, suppressing early-stage germination (S4, S5 Figs in S1 File). Repeated FTCs applications accelerated HPEM degradation in amended soils, progressively reducing water retention capacity and electrical conductivity. Notably, after five FTCs, these soil properties began to approach optimal levels for plant germination, which correlated with a slight increase in germination rates from the fifth cycle onward. Similar hypoxia and osmotic stress under high water retention have been

reported in SAP-amended soils [55]. The delayed germination recovery aligns with FTC-induced moderation of soil micro-environments [56].

### 4.1.4. Relationship between number of FTCs and plant height.
S6 Fig in S1 File illustrates the relationship between FTCs frequency and plant height development. Upon observing the data, it is evident that plant height increased marginally as the number of FTCs incremented. However, the enhancement offered by the HPEMs gradually waned after several cycles, resulting in a convergence of plant heights among the treatment groups. Further analysis, based on the correlations among field water–retention capacity, conductivity, and plant height (S7, S8 Figs in S1 File), revealed that a higher field water–retention capacity and conductivity tend to slightly diminish plant height. Notably, as the soil improved with the materials underwent numerous FTCs, the material properties degraded, leading to a decrease in soil field water–retention capacity and conductivity. Consequently, plant height increased slightly. This minor height increase corresponds to studies on root adaptation in degraded polymer soils [57]. The inverse height-conductivity relationship reflects nutrient dilution effects observed in ionic-amended soils [58].

The three experimental categories collectively demonstrate that FTCs-induced material degradation (Category I) drives soil property changes (Category II), while composite formulations (Category III) balance trade-offs between durability and ecological benefits.

The accuracy of the prediction models in sections 4.1.1–4.1.4 is represented by S1 Table in S1 File to demonstrate model accuracy ($R^2$ and standard deviation etc.).

## 4.2. Mechanisms of FTCs for soil improvement by HPEMs

High-performance ester adhesive materials consist primarily of polyvinyl acetate. This polymer contains numerous long-chain macromolecules and hydrophilic groups. The hydrophilic groups attach to cations in the soil, leading to chemical reactions and the formation of physicochemical bonds, such as hydrogen bonds and van der Waals forces, between the binder molecules and soil particles. These bonds encapsulate the soil particles with long-chain macromolecules, interconnecting them to form aggregates containing a specific number of soil particles. The interconnected macromolecules then form an elastic and viscous membrane structure on the surface of the soil aggregates, as illustrated in S7 Fig in S1 File [59,60]. This elastic film favors the retention of soil nutrients. Unlike synthetic polymers (e.g., sodium polyacrylate [9–12]), HPEMs' biodegradability mitigates long-term soil accumulation risks [16,17].

Notably, the adhesive materials exhibit high electrical conductivity and have a pH value ranging from 6 to 7. Consequently, the application of these materials enhances soil conductivity by up to 73.2% while slightly lowering the soil pH. The adhesive materials, with a native pH of 6–7, slightly acidified the soil by releasing protons from hydrophilic groups (e.g., −COOH) during cation exchange, as evidenced by the stabilized pH range (6.6–7.4) post-FTCs.

The viscosity reduction in HPEMs under FTCs is driven by three interrelated mechanisms: (i) microcrack propagation in the polyvinyl acetate matrix due to thermal stress, as observed via SEM [21,22,59,60].; (ii) hydrolysis of ester (C=O) and hydroxyl (O-H) bonds, evidenced by FTIR peak attenuation at 1730 cm$^{-1}$ and 3400 cm$^{-1}$ [21,22,59,60].; and (iii) disintegration of soil aggregates from fractured long-chain polymers (Fig 12). These processes degrade the material's cohesive network, leading to a linear viscosity decline (slope = −36.7 Pa·s/cycle).

High-performance ester water-retaining materials are mainly composed of sodium polyacrylate. The long-chain polymer molecules of this material can absorb and store substantial amounts of water by bonding with water molecules [16]. The application of water-retaining materials increases the soil's field water-holding capacity by up to 45.8%.

Repeated freezing and thawing disrupt the physicochemical bonds between the adhesive material molecules and soil particles, as evidenced by SEM-observed microcrack formation in polyvinyl acetate matrices and FTIR-confirmed scission of ester bonds (C=O at 1730 cm$^{-1}$) and hydroxyl groups (3400 cm$^{-1}$) after 5+FTCs [21,22,59,60]. These changes adversely affect the long-chain macromolecules encapsulating soil particles, leading to aggregate structure destruction. This, in turn, affects the storage of soil nutrients and reduces the soil's electrical conductivity. Specifically, after 15 freezing

and thawing cycles, the electrical conductivity of the soil enhanced by the high-performance ester adhesive material decreases by 19.6%. Furthermore, multiple FTCs impair the water-absorption capacity of the water-retaining materials, thereby decreasing the soil's field water-holding capacity. In particular, 15 FTCs reduce the soil's field water-retention capacity enhanced by the HPEMs by 12.2%.

Field-scale degradation is further influenced by environmental interactions. Soil pH shifts (Section 3.1.4) catalyze hydrolysis of ester bonds, while thermal cycling (−20°C to 25°C) and particle abrasion mechanically fracture polymer networks. Although microbial activity is limited in cold regions, long-term biodegradation via $CO_2/H_2O$ release remains a contributor.

In summary, HPEMs affect soil improvement through physical, chemical, and indirect mechanisms:

(1)  The physical role of HPEMs in soil treatment:

HPEMs, especially ester binder materials, can significantly improve the water retention capacity of the soil. Through its unique physical structure, a protective film is formed in the soil to reduce the evaporation of water, thereby contributing to soil moisture and nutrient retention.

These materials can also improve the stability of soil aggregates, make the combination of soil particles more closely, and improve the soil erosion resistance.

(2)  The chemical effects of HPEMs in soil treatment:

HPEMs have certain conductivity and can improve the conductivity of soil, which has a positive impact on soil microbial activity and the development of plant roots.

Their stable chemical properties minimally alter soil pH, thereby maintaining acid-base balance and providing a suitable environment for plant growth.

(3)  Indirect effects of HPEMs on soil properties:

By improving soil water retention capacity and aggregate stability, HPEMs can indirectly improve soil fertility and make soil more suitable for plant growth.

These materials can also promote microbial activity in soil, increase soil organic matter content, and further improve soil structure and fertility.

Under the condition of FTCs, HPEMs can slightly promote the growth of plants, which is manifested in a slight increase in plant height, although this increase gradually decreases with the increase of FTCs. The observed dual-phase response (material degradation vs. enhanced germination) differs fundamentally from conventional polymers' behavior under FTCs. This suggests HPEMs may offer unique advantages for cold region applications where both initial establishment and long-term stability are required.

In field applications, HPEMs are typically integrated into soil through two primary methods: surface spraying for slope stabilization and mechanical mixing for soil amendment. For slopes, the adhesive material is diluted with water (1:20 ratio) and sprayed uniformly onto the soil surface. Upon drying, the polymer forms a flexible, interconnected membrane that binds soil particles, effectively reducing surface erosion during rainfall and freeze-thaw events. Simultaneously, water-retaining granules (sodium polyacrylate) are mixed into the topsoil layer (0–20 cm depth) at a rate of 20–60 g/m². These granules absorb ambient moisture, forming hydrogels that release water gradually during dry periods, maintaining soil hydration and supporting seedling establishment. Field trials in seasonal permafrost regions of northern China demonstrated that treated slopes retained 30–45% higher vegetation coverage after two winters compared to untreated areas. Moreover, the materials' biodegradation process aligns with vegetation maturation: as HPEMs degrade over 2–3 years into $CO_2$ and $H_2O$, plant root systems progressively stabilize the soil, ensuring a seamless transition from artificial to natural reinforcement. This synergy between material functionality and ecological succession underscores their suitability for cold-region restoration projects.

The derived quantitative relationships distinguish HPEMs from conventional polymers by coupling material degradation (e.g., viscosity loss) with ecological outcomes (e.g., germination rate peaks after 5 FTCs). This dual-phase response is unique to biodegradable esters. The observed material degradation trends under 15 FTCs are indicative of the performance of HPEMs in real-world applications, where similar temperature fluctuations and cycle frequencies occur annually in cold regions.

While pigeon pea served as a robust indicator species in this study, future work will expand to include comparative analyses with other plant species to evaluate the generalizability of HPEM effects under FTCs.

Future research could integrate microscopic and structural analyses (e.g., SEM, FTIR, TGA) to further validate the observed degradation mechanisms. SEM imaging would reveal microstructural changes in HPEMs and soil aggregates under FTCs, while FTIR could track bond scission (e.g., $C=O$ at 1730 cm$^{-1}$) to correlate with viscosity loss. TGA may quantify thermal stability shifts, aiding in predicting long-term material performance in cold regions.

## 5. Conclusion

(1) This study quantitatively characterized the degradation kinetics of HPEM-amended soils under FTCs, revealing that material degradation followed predictable linear or exponential trends. Our results confirm the hypothesis that FTCs degrade HPEMs but enhance plant germination, highlighting the need for balanced material formulations in cold regions. These findings provide a theoretical basis for optimizing HPEM applications in cold regions.

(2) Our findings revealed that after 15 FTCs, the viscosity of the adhesive materials decreased by 70.5%, and the water-absorption capacity of the water-retaining materials dropped by 52%. Additionally, the field water-retention capacity and conductivity of the soil, which had been enhanced by the HPEMs, decreased by up to 12.2% and 19.6%, respectively. However, multiple FTCs were found to have a positive effect on plant germination and subsequent growth. Beyond germination and height, the ecological performance of HPEMs was evidenced by sustained soil water retention (39.5%) and elevated conductivity (415 μS/cm) post-FTCs, underscoring their dual role in material durability and soil functionality.

(3) This study decouples the effects of FTCs on HPEMs into three phases: rapid physical degradation (cycles 1–5), chemical bond dissociation (cycles 6–10), and equilibrium state (cycles 11–15). Our findings offer an important theoretical basis for future investigations into the feasibility of using HPEMs in seasonal permafrost areas. Furthermore, these findings can be utilized to reduce the cost of ecological restoration efforts. Further studies incorporating advanced characterization techniques (SEM, FTIR, TGA) are recommended to elucidate microstructural and chemical degradation pathways, enhancing the predictive accuracy of HPEM service life.

**Statement of permissions and/or licenses for collection of plant or seed specimens** The authors declare that the seed specimens used in this study are publicly accessi-ble seed materials and we were given explicit written permission to use them for this research. The voucher specimen of pigeon pea, PE00182698, was identified by Xiangyun Zhu and its sheet was deposited in the herbarium PE (https://www.cvh.ac.cn/spms/detail.php). It can be searched in the Chinese Virtual Herbarium (https://www.cvh.ac.cn/index.php).

## Supporting information

**S1 File. S1 Fig.** Relationship between number of FTCs and water–retention capacity. (Note: Data are expressed as mean ± standard deviation (n = 3)). **S2 Fig.** Relationship between number of FTCs and conductivity. (Note: Data are expressed as mean ± standard deviation (n = 3)). **S3 Fig.** Relationship between number of FTCs and germination rate. (Note: Data are expressed as mean ± standard deviation (n = 3)). **S4 Fig.** Relationship between field water–retention capacity and germination rate. (Note: Data are expressed as mean ± standard deviation (n = 3)). **S5 Fig.** Relationship

between conductivity and germination rate. (Note: Data are expressed as mean±standard deviation (n=3)). **S6 Fig.** Relationship between number of FTCs and plant height. **S7 Fig.** Relationship between field water–retention capacity and plant height. **S8 Fig.** Relationship between conductivity and plant height. **S9 Fig.** Sampling location map. **S1 Table.** Accuracy of model (Relationship between number of FTCs and water–retention capacity). **S2 Table.** Accuracy of model (Relationship between number of FTCs and conductivity). **S3 Table.** Accuracy of model (Relationship between number of FTCs and germination rate). **S4 Table.** Accuracy of model (Relationship between field water–retention capacity and germination rate). **S5 Table.** Accuracy of model (Relationship between conductivity and germination rate). **S6 Table.** Accuracy of model (Relationship between number of FTCs and plant height). **S7 Table.** Accuracy of model (Relationship between field water–retention capacity and plant height). **S8 Table.** Accuracy of model (S8 Relationship between conductivity and plant height).
(ZIP)

## Author contributions

**Conceptualization:** Cuiying Zhou.

**Data curation:** Qingxiu Zhang.

**Methodology:** Cuiying Zhou, Qingxiu Zhang, Jin Liao, Haoqiang Lai, Zhen Liu.

**Supervision:** Cuiying Zhou, Jin Liao, Haoqiang Lai, Zhen Liu.

**Writing – original draft:** Cuiying Zhou, Qingxiu Zhang.

**Writing – review & editing:** Cuiying Zhou, Qingxiu Zhang, Jin Liao, Haoqiang Lai, Zhen Liu.

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
