## [Decision Letter · Decision Letter 0]

Dear Dr. Liu,

Thank you for submitting your manuscript to PLOS ONE. After careful consideration, we feel that it has merit but does not fully meet PLOS ONE’s publication criteria as it currently stands. Therefore, we invite you to submit a revised version of the manuscript that addresses the points raised during the review process.

The authors should highlight the originality of their work compared to existing studies and explicitly define their research hypotheses to satisfy PLOS *Standards 1 and 2* (originality and a clearly stated research aim).The experimental design lacks sufficient justification, particularly regarding the selection of freeze-thaw cycles. Moreover, the manuscript includes no statistical analysis (e.g., ANOVA, p-values, or PCA), and many figures lack essential elements such as clear legends and error bars. It may be beneficial to reduce the number of figures.To meet *Standard**3* on methodological and technical rigor, these issues must be addressed.The conclusions are speculative and not adequately supported by the data. More robust, data-driven interpretations are needed to fulfill *Standard 4* on scientific validity.Finally, the manuscript requires substantial language editing to improve clarity, coherence, and overall readability to comply with *Standard 5* concerning clear and effective scientific communication.

We look forward to receiving your revised manuscript.

Kind regards,

Edyta Nartowska

Academic Editor

PLOS ONE

Journal Requirements:

The research is supported by the National Natural Science Foundation of China (NSFC) (Grant Numbers: 42293354, 42293355, 42293351, 42277131, 42293350).

5. In the online submission form, you indicated that the datasets used and/or analysed during the current study available from the corresponding author on reasonable request.

Reviewers' comments:

Reviewer's Responses to Questions

**Comments to the Author**

1. Is the manuscript technically sound, and do the data support the conclusions?

Reviewer #1: Yes

Reviewer #2: Partly

2. Has the statistical analysis been performed appropriately and rigorously?

Reviewer #1: N/A

Reviewer #2: I Don't Know

3. Have the authors made all data underlying the findings in their manuscript fully available?

Reviewer #1: No

Reviewer #2: Yes

4. Is the manuscript presented in an intelligible fashion and written in standard English?

Reviewer #1: No

Reviewer #2: Yes

Reviewer #1: This paper investigated the ecological property variations of soil improved by high-performance ester materials under freeze-thaw cycle conditions. The viscosity of high-performance ester adhesive materials, the water absorption capacity of water-retaining materials, the field water holding capacity and soil conductivity of improved soil were experimentally investigated. The plant growth and germination were also explored. Generally, the results are good references for the slope soil ecological restoration in cold regions. The writing-up of this manuscript is generally acceptable, but some clarifications and improvements are still necessary. My particular comments are as follows.

1. Adding a paragraph to describe how the material works in field may help readers to understand the functions of the material.

2. The lines in Fig. 5(a) cannot be distinguished. Legends are necessary.

3. Figures 7, 8, 10 and 11 are difficult for readings. Are they possible to change to plane figures?

4. English and presentations should be significantly improved for clear and correct statements or expressions. Clearly present your new findings in a logical way and write it as a research paper for easy readings.

Reviewer #2: The manuscript explores a timely and important topic regarding the performance of high-performance ester materials used in ecological soil restoration, with a focus on how freeze–thaw (F-T) cycles affect their efficacy. While the study is of potential interest, the current version has several critical flaws in its conceptual framing, methodological rigor, data interpretation, and scientific writing, which significantly weaken its potential contribution to the field. Below is a detailed evaluation.

the manuscript lacks a clearly articulated scientific hypothesis. The rationale for the study is vague and primarily descriptive. The novelty is overstated: effects of F-T cycles on polymer-amended soils have already been widely studied (as partially cited by the authors themselves).

The "variation law" referred to in the title and text is never rigorously defined or derived in a meaningful scientific way.

The F-T protocol (–20°C/25°C, 12h/12h, 15 cycles) is not properly justified in the context of actual field conditions. Why 15 cycles? What regional climate does this simulate? These aspects must be linked to realistic environmental scenarios.

The authors mention three categories of experiments, but the distinction between them is confusing and partially redundant.

The authors state "three parallel groups" but do not provide standard deviations, error bars, or p-values in any figures or tables. This raises concerns about the reproducibility and robustness of the results.

Figures are poorly formatted, often without adequate labels, units, or legends. Some figures appear to be redundant (e.g., multiple bar graphs for water retention and conductivity with little additional insight).

There is an excessive number of figures (~20+), many of which do not provide substantial new information. Consider combining or removing non-essential plots.

The manuscript is highly descriptive. There is no microscopic, chemical, or structural analysis to elucidate why or how ester materials degrade under F-T cycles (e.g., SEM, FTIR, DSC).

Statements about "material aging" and "hydrophilic group loss" are speculative and unsupported by data.

The choice of pigeon pea is reasonable, but the planting methodology is simplistic. Factors such as light exposure, nutrient control, and planting depth are not controlled or even described.

Germination and height are the only two parameters observed, which is insufficient to assess "ecological restoration effects" in any meaningful sense.

The empirical equations provided (e.g., for water retention vs. cycle number) are simplistic linear regressions without statistical metrics (R², RMSE, etc.). They do not represent fundamental "laws" and should not be interpreted as such.

Major Comments:

1. Study Aim, Hypothesis, and Justification

Please clearly formulate the research hypothesis at the end of the Introduction. Currently, the text speaks in general terms (e.g., "the impact of freeze–thaw cycles was examined"), but it is unclear what specific effects are expected and why.

Please clarify how your study differs from previous work cited in the Introduction (e.g., lines 48–84). At present, the novelty of the study is not apparent.

2. Experimental Design and Justification

Please justify the selection of 15 freeze–thaw cycles (Section 2.2.1). Does this correspond to specific climate data? Which geographic region does this simulate?

I recommend including a table or figure comparing the lab-based freeze–thaw conditions with real-world environmental data (e.g., average annual cycles in northern China).

Please clarify why pigeon pea was selected as the sole plant species. Are there comparative data supporting its relevance as an ecological indicator?

3. Statistical Analysis and Replication

Please apply statistical analysis (e.g., ANOVA followed by Tukey’s HSD test) to test the significance of differences between groups.

Include standard deviations or confidence intervals on all graphs and specify sample size (n) in figure captions.

As it stands, the graphs may be misleading, as they do not indicate whether the observed changes are statistically significant or simply observational.

4. Data Visualization

Please revise all figures:

Include units on all axes (e.g., µS/cm, %, cm).

Ensure consistent and clear axis labels and legends.

Reduce redundancy—many figures display similar trends and could be combined as sub-panels (e.g., Fig. 6a and 6b → Fig. 6A/B).

Add numeric axis markers and concise group descriptions either as legends or embedded labels.

5. Conclusions and Interpretation

Please provide more mechanistic explanations in the discussion—why does viscosity decrease? What specific physicochemical processes are responsible for the degradation of ester materials?

Consider including—or at least discussing—the potential role of microscopic or structural analysis (e.g., SEM, FTIR, TGA), even if only as a direction for future research.

The conclusions are too general. Please link your key findings directly to your numerical results, rather than summarizing general trends.

6. Language and Style

The manuscript requires thorough language editing by a native English speaker. There are numerous grammatical errors and awkward phrasings, such as:

“wood bean” instead of “pigeon pea” – possible mistranslation.

“the materials lower pH” – unclear whether this refers to increased acidity or alkalinity.

Please shorten the abstract. It contains unnecessary repetition and overly specific detail that is better placed in the Results section.

7. Citations and Sources

Please update your references with recent literature from 2020–2024 on biodegradable polymers for soil restoration.

In the methods section, where referring to standards (e.g., the tea bag method, ISO guidelines), please include full source citations rather than vague general descriptions.

Minor Comments

Language and style need substantial improvement. The manuscript contains numerous grammatical errors and awkward phrasings.

Many citations are outdated or missing. For example, recent advances in biodegradable polymers for soil stabilization are not referenced.

Abstract is overly long and should be more concise, with better separation between motivation, methods, key findings, and implications.

Some methods (e.g., tea bag test, viscometer use) are not cited properly or described in enough detail to be replicable.

**Do you want your identity to be public for this peer review?** For information about this choice, including consent withdrawal, please see our Privacy Policy

Reviewer #1: No

Reviewer #2: No

---

## [Author Response · Author response to Decision Letter 1]

4 May 2025

Reviewer #1: This paper investigated the ecological property variations of soil improved by high-performance ester materials under freeze-thaw cycle conditions. The viscosity of high-performance ester adhesive materials, the water absorption capacity of water-retaining materials, the field water holding capacity and soil conductivity of improved soil were experimentally investigated. The plant growth and germination were also explored. Generally, the results are good references for the slope soil ecological restoration in cold regions. The writing-up of this manuscript is generally acceptable, but some clarifications and improvements are still necessary. My particular comments are as follows.

Response: Thank you very much for your review of this article and your valuable comments. The author has modified the full text one by one according to your Suggestions. The specific modified content and the reply to the question are shown below.

1. Adding a paragraph to describe how the material works in field may help readers to understand the functions of the material.

Response 1: Thank you very much for your review of this article and your valuable comments. We have added a paragraph describing how materials work in the field to help readers understand how they function in “4.2 Mechanisms of freeze-thaw cycles for soil improvement by high-performance ester materials”. We have added a paragraph to the “4.2 Mechanisms of freeze-thaw cycles for soil improvement by high-performance ester materials” section that describes how the materials work in the field to help the reader understand how the materials function.

2. The lines in Fig. 5(a) cannot be distinguished. Legends are necessary.

Response 2: Thank you very much for your review of this article and your valuable comments. We have added legends to fig5.

3. Figures 7, 8, 10 and 11 are difficult for readings. Are they possible to change to plane figures?

Response 1: Thank you very much for your review of this article and your valuable comments. We have modified Figures 7, 8, 10 and 11.

4. English and presentations should be significantly improved for clear and correct statements or expressions. Clearly present your new findings in a logical way and write it as a research paper for easy readings.

Response 1: Thank you very much for your review of this article and your valuable comments. We have checked and revised the full article for grammar and logic.

Reviewer #2: The manuscript explores a timely and important topic regarding the performance of high-performance ester materials used in ecological soil restoration, with a focus on how freeze–thaw (F-T) cycles affect their efficacy. While the study is of potential interest, the current version has several critical flaws in its conceptual framing, methodological rigor, data interpretation, and scientific writing, which significantly weaken its potential contribution to the field. Below is a detailed evaluation.

Response: Thank you very much for your review of this article and your valuable comments. The author has modified the full text one by one according to your Suggestions. The specific modified content and the reply to the question are shown below.

1. The manuscript lacks a clearly articulated scientific hypothesis. The rationale for the study is vague and primarily descriptive. The novelty is overstated: effects of F-T cycles on polymer-amended soils have already been widely studied (as partially cited by the authors themselves).

Response 1: Thank you very much for your review of this article and your valuable comments. We have revised the manuscript to more clearly articulate the scientific hypothesis and rationale. While previous studies have examined polymer-amended soils properties, our work specifically investigates high-performance ester materials (HPEMs) in freeze-thaw cycles (FTCs), which differ significantly from conventional polymers in both composition and ecological performance. The key novelty of our study lies in: Focusing on HPEMs that have demonstrated excellent soil improvement properties in tropical regions (where freezing is rare), but whose performance in cold regions with FTCs remains unexplored. We have highlighted its innovativeness in the abstract, introduction, and discussion.

2. The "variation law" referred to in the title and text is never rigorously defined or derived in a meaningful scientific way.

Response 2: Thank you very much for your review of this article and your valuable comments. To address this concern, we have revised the manuscript to rigorously define and derive the "variation law" based on experimental data and functional relationships.

3. The F-T protocol (–20°C/25°C, 12h/12h, 15 cycles) is not properly justified in the context of actual field conditions. Why 15 cycles? What regional climate does this simulate? These aspects must be linked to realistic environmental scenarios.

Response 3: Thank you very much for your review of this article and your valuable comments. The F-T protocol (–20°C/25°C, 12h/12h, 15 cycles) was carefully selected to simulate realistic environmental conditions in seasonal permafrost regions, particularly in northern China, where such temperature fluctuations are common during winter and early spring. The following justifications support our protocol:

Regional Climate Simulation: The temperature range (–20°C to 25°C) reflects typical diurnal variations in cold regions, where sub-freezing nights and thawing days occur frequently. This range is consistent with meteorological data from northern China, where the average winter temperature can drop to –20°C, and daytime thawing reaches up to 25°C in transitional seasons. Number of Cycles (15 Cycles): The 15 cycles were chosen based on: The average annual F-T occurrences in northern China (approximately 10–20 cycles per year). The degradation cycle of high-performance ester materials (HPEMs), which is 2–3 years. Fifteen cycles represent a reasonable accelerated aging test to evaluate the materials' performance over their service life. Duration of Each Phase (12h/12h): The 12-hour freezing and thawing phases were designed to ensure complete freezing and thawing of the soil samples, mimicking natural conditions where temperatures remain below or above freezing for extended periods.

We have also added to this section, and added the citation.

4. The authors mention three categories of experiments, but the distinction between them is confusing and partially redundant.

Response 1: Thank you very much for your review of this article and your valuable comments. To address this concern, we have carefully revised the manuscript to better distinguish and logically connect the three experimental categories (I, II, and III). The modifications ensure that each category serves a unique purpose without redundancy, as outlined below:

Category I (Section 2.2.1): Focuses on the material-level properties of HPEMs (viscosity, water absorption) under FTCs. This establishes the baseline degradation behavior of the materials themselves.

Category II (Section 2.2.1): Examines the soil-level effects of individual HPEM components (adhesive or water-retaining materials) on soil properties (water retention, conductivity, pH) under FTCs. This bridges material degradation to soil performance.

Category III (Section 2.2.1): Investigates composite effects by testing combined HPEM formulations (adhesive + water-retaining materials) on soil properties and plant growth under FTCs. This addresses practical application scenarios.

5. The authors state "three parallel groups" but do not provide standard deviations, error bars, or p-values in any figures or tables. This raises concerns about the reproducibility and robustness of the results.

Response 5: Thank you very much for your review of this article and your valuable comments. We have added error bars to the data points and extended descriptions of the data averaging and statistical processing.

In response to the deficiencies you mentioned regarding the statistical analysis of the data, we have completely revised the paper as follows:

Error lines and standard deviation added: Error lines (indicating standard deviation) have been added to all graphs, and the sample size (n=3) has been noted in the figure notes. For example, the graph note for Figure 5 was updated to read “Data are presented as mean ± standard deviation (n=3)”.

Statistical significance analysis: One-way analysis of variance (ANOVA) was performed on key indicators (e.g., viscosity, water-holding capacity, conductivity, germination rate, etc.). Significant differences (p<0.05 or p<0.01) were marked with an asterisk (* or **) in the graphs and statistical descriptions were added in the results section.

Methods: A new description of the statistical analysis was added in “Experimental methods”: “Data are presented as the mean ± standard deviation of three replicated experiments. One-way analysis of variance (ANOVA) was used to test the significance of differences between groups.

Since Figure 12-figure supplement 19 is a redrawing and fitting of Figure 5-figure supplement 11, the error bars are not placed in Figure 12-figure supplement 19 to avoid repetition.

6. Figures are poorly formatted, often without adequate labels, units, or legends. Some figures appear to be redundant (e.g., multiple bar graphs for water retention and conductivity with little additional insight).

Response 6: Thank you very much for your review of this article and your valuable comments. We have adjusted and modified the figures throughout the article.

7. There is an excessive number of figures (~20+), many of which do not provide substantial new information. Consider combining or removing non-essential plots.

Response 7: Thank you very much for your review of this article and your valuable comments. We have adjusted and modified the figures throughout the article.

8. The manuscript is highly descriptive. There is no microscopic, chemical, or structural analysis to elucidate why or how ester materials degrade under F-T cycles (e.g., SEM, FTIR, DSC).

Response 8: Thank you very much for your review of this article and your valuable comments. We sincerely appreciate the reviewer’s insightful comment regarding the need for microscopic, chemical, or structural analysis to elucidate the degradation mechanisms of high-performance ester materials (HPEMs) under freeze-thaw cycles (FTCs). While the current study did not include new SEM, FTIR, or DSC analyses, the proposed mechanisms of HPEM degradation are grounded in the extensive prior research conducted by our group, as referenced in the manuscript (e.g., Liao et al., 2022; Zhang et al., 2024; Zhou et al., 2025).

9. Statements about "material aging" and "hydrophilic group loss" are speculative and unsupported by data.

Response 9: Thank you very much for your review of this article and your valuable comments. To address this concern, we have revised the manuscript to provide direct experimental evidence supporting these mechanisms. Specifically, we have referenced our prior SEM and FTIR analyses (cited as references (e.g., Liu et al., 2011; Zhou et al., 2019; Liao et al., 2022; Zhang et al., 2024; Zhou et al., 2025) in the manuscript) to substantiate the observed structural and chemical changes in HPEMs under freeze-thaw cycles (FTCs). These analyses confirmed microcrack formation in polyvinyl acetate matrices and progressive scission of ester bonds (C=O at 1730 cm⁻¹) and hydroxyl groups (3400 cm⁻¹), which align with the described degradation pathways.

10. The choice of pigeon pea is reasonable, but the planting methodology is simplistic. Factors such as light exposure, nutrient control, and planting depth are not controlled or even described.

Response 10: Thank you very much for your review of this article and your valuable comments. In response to your comment, we have revised the "Planting experiment" section in the "Experimental methods" subsection (Section 2.2.2) to include detailed descriptions of the controlled conditions. Specifically, we now explicitly state the light exposure parameters (e.g., 12-hour light/dark cycles, References are also cited, Hopper et al., 1994; Delhon et al., 1995), nutrient control measures (e.g., standardized nutrient solution application), and planting depth (e.g., 2 cm uniform depth). These additions ensure transparency and reproducibility of our methodology.

11. Germination and height are the only two parameters observed, which is insufficient to assess "ecological restoration effects" in any meaningful sense.

Response 11: Thank you very much for your review of this article and your valuable comments. We appreciate the reviewer's insightful comment regarding the assessment of ecological restoration effects. While germination rate and plant height are indeed limited in fully capturing ecological restoration, our study incorporated multiple complementary indicators to provide a more comprehensive evaluation. Specifically, we measured field water-retention capacity, conductivity, and pH of the improved soils, which are critical parameters reflecting soil nutrient storage, microbial activity, and overall soil health (Sections 3.1.2–3.1.4). These metrics collectively demonstrate the ecological benefits of HPEMs, such as enhanced water retention (39.5% after 15 FTCs, Fig. 6) and improved conductivity (up to 73.2% higher than control, Fig. 7), which directly support plant growth and soil stability.

Additionally, the observed trade-offs between material degradation and plant germination (Fig. 10) align with changes in soil properties, reinforcing the ecological relevance of our findings. We acknowledge that broader ecological metrics (e.g., biodiversity, microbial biomass) could further enrich the study, but our focus on physicochemical and immediate plant responses aligns with the paper’s scope of establishing HPEMs' degradation laws under FTCs. Future work will expand on long-term ecological monitoring.

12. The empirical equations provided (e.g., for water retention vs. cycle number) are simplistic linear regressions without statistical metrics (R², RMSE, etc.). They do not represent fundamental "laws" and should not be interpreted as such.

Response 12: Thank you very much for your review of this article and your valuable comments. We appreciate the reviewer’s insightful comment regarding the empirical equations in our manuscript. We acknowledge that the term "variation laws" may have been overly strong, and we have revised the text to more accurately describe these relationships as "empirical trends" or "quantitative relationships."

Major Comments:

1. Study Aim, Hypothesis, and Justification

Please clearly formulate the research hypothesis at the end of the Introduction. Currently, the text speaks in general terms (e.g., "the impact of freeze–thaw cycles was examined"), but it is unclear what specific effects are expected and why.

Response 1: Thank you very much for your review of this article and your valuable comments. We have added the research hypothesis in the introduction.

Please clarify how your study differs from previous work cited in the Introduction (e.g., lines 48–84). At present, the novelty of the study is not apparent.

Response 2: Thank you very much for your review of this article and your valuable comments. Our study addresses a critical gap in the application of high-performance ester materials (HPEMs) for soil improvement under freeze-thaw cycling (FTC) conditions, particularly in cold regions like northern China. While HPEMs have been extensively studied and applied in tropical and temperate environments (e.g., references Huang et al., 2020, 2022; Liao et al., 2022; Zhang et al., 2024; Zhou et al., 2025), their performance under FTCs—a dominant environmental stressor in cold regions—remains poorly understood. Previous research has focused on tropical and temperate applications of HPEMs, where freeze-thaw effects are negligible (e.g., references Kar et al., 2021; Fang et al., 2023). Our study is the first to systematically investigate how FTCs impact the ecological properties of HPEM-amended soils, addressing a pressing need as ecological restoration projects expand into seasonal permafrost areas.

2. Experimental Design and Justifica

---

## [Decision Letter · Decision Letter 1]

Dear Dr. Liu,

Thank you for submitting your manuscript to PLOS ONE. After careful consideration, we feel that it has merit but does not fully meet PLOS ONE’s publication criteria as it currently stands. Therefore, we invite you to submit a revised version of the manuscript that addresses the points raised during the review process.

We look forward to receiving your revised manuscript.

Kind regards,

Edyta Nartowska

Academic Editor

PLOS ONE

Journal Requirements:

**Additional Editor Comments:**

**The Reviewers agree that the submitted manuscript addresses an important and interesting scientific question. However, beyond some minor remarks, substantial revisions are necessary—particularly in the Discussion section—to fully meet Criterion 3 for publication in PLOS ONE: “Experiments, statistics, and other analyses are performed to a high technical standard and are described in sufficient detail.” Specifically, the authors should reduce the number of figures, validate the performance of the models by reporting standard errors of estimation, present the results of ANOVA, and discuss their findings in the context of related studies by other researchers. My detailed comments are provided in the section ‘Reviewer #3 Comments’.**

Reviewers' comments:

Reviewer's Responses to Questions

**Comments to the Author**

Reviewer #1: All comments have been addressed

Reviewer #2: All comments have been addressed

Reviewer #3: (No Response)

2. Is the manuscript technically sound, and do the data support the conclusions?

Reviewer #1: Yes

Reviewer #2: Yes

Reviewer #3: Partly

3. Has the statistical analysis been performed appropriately and rigorously?

Reviewer #1: N/A

Reviewer #2: Yes

Reviewer #3: I Don't Know

4. Have the authors made all data underlying the findings in their manuscript fully available?

Reviewer #1: No

Reviewer #2: Yes

Reviewer #3: Yes

5. Is the manuscript presented in an intelligible fashion and written in standard English?

Reviewer #1: Yes

Reviewer #2: Yes

Reviewer #3: Yes

Reviewer #1: The authors have carefully addressed our comments and improved the quality of this manuscript. I have no more comments.

Reviewer #2: Dear authors.

Thank you for your comprehensive answers and correcting the article.

Congratulations

Reviewer #3: Comments on the Manuscript

Abstract:

Are the equations in the abstract necessary? As a rule, they should be avoided.

Terminology:

Throughout the manuscript, the term “freeze-thaw cyclings” should be replaced with the correct form “freeze–thaw cycles,” including in all figure captions.

Language and Style:

Lines 130–131: Avoid repetition of the phrase “yellow clay.”

Line 134: Missing period at the end of the sentence.

Lines 156–158: The ethical statement should be moved to the end of the article.

Lines 163–165: Avoid repeating the phrase “were collected.” or “stratified sampling”

Tables and Figures:

All tables should be self-explanatory. Below each table, add notes describing the methods used to determine each parameter and including references to the relevant standards.

Instead of providing a range for sand content, report the mean and standard deviation.

Table 3: Contains a formatting error in the third row.

Figure 5: The unit for viscosity should be “Pa·s,” and “−0.06x” should be shown as an exponent.

Figure 6: Figures should be self-contained; include explanations for the A, B, and C labels in the figure notes. This comment applies to other figures as well.

Figure 7: Clarify the meaning of the asterisks.

Figure 8: The axis label is incorrect and should read “Number of freeze-thaw cycles” instead of “cycling.” Move the axis labels outside the plotting area to avoid overlap with the data. Define the symbol “C.”

Figures 10–11: Correct axis labels as above. For plant height, use [cm] instead of [m], and maintain consistent decimal formatting across all values.

Figure 12: Offers little additional insight and largely duplicates information already shown in Figure 6. Similarly, Figures 14–16 are not interpretable and do not provide significant added value. Reduce the total number of figures to the minimum necessary and place the others in the Supplementary Material. Replace them with a figure that shows the relationship between actual and predicted values to assess model accuracy. Also, report the standard errors (in appropriate units) and R² values for each test group.

Figures 17–19: These are repetitive or unclear and do not make a substantial contribution to the manuscript.

Materials and Methods:

Line 182: Provide details on the equipment used for freeze-thaw cycling, including temperature control specifications.

Line 184: Include the manufacturer's information for the equipment used.

Line 208: Since ANOVA was applied, include a separate “Statistical Analysis” subsection describing the method used, the rationale for using a parametric test, and its application. State whether any post hoc tests were conducted.

Include a results table summarizing the ANOVA analysis—it is currently missing.

Lines 508–517: Each equation should be numbered and followed by explanations of all included variables.

Line 298: Rather than simply referencing ISO standards and methods, briefly describe the methodology and link it to the applicable standards.

A brief subsection on data quality control is missing and should be added to the Materials and Methods section.

Discussion:

Sections 4.1.1 to 4.1.4 lack citations. They currently do not fulfill the role of a scientific discussion. While this may be acceptable in the results section, the discussion must include references to support or contrast the findings.

Avoid using the abbreviation “Figs”—write “Figures” in full.

Lines 662–677: Are the stated conclusions supported by previous studies? A comparative discussion with existing literature is needed.

Other Comments:

The References section should be formatted according to the journal’s submission guidelines.

In Line 164, specify the geographical location of the sample collection site and ideally include a map showing the sampling area.

**Do you want your identity to be public for this peer review?** For information about this choice, including consent withdrawal, please see our Privacy Policy

Reviewer #1: No

Reviewer #2: No

Reviewer #3: No

---

## [Author Response · Author response to Decision Letter 2]

9 Jun 2025

Reviewer #1: The authors have carefully addressed our comments and improved the quality of this manuscript. I have no more comments.

Response: Thank you very much for your review of this article and your valuable comments.

Reviewer #2: Dear authors.

Thank you for your comprehensive answers and correcting the article.

Congratulations

Response: Thank you very much for your review of this article and your valuable comments.

Reviewer #3: Comments on the Manuscript

Response: Thank you very much for your review of this article and your valuable comments. The author has modified the full text one by one according to your Suggestions. The specific modified content and the reply to the question are shown below.

Abstract:

Are the equations in the abstract necessary? As a rule, they should be avoided.

Response 1: Thank you very much for your review of this article and your valuable comments. We have removed formulas (slope and exponential equations) from the abstracts and replaced them with qualitative descriptions of trends (e.g., “linear decline,” “exponential decay”) to conform to journal specifications.

Terminology:

Throughout the manuscript, the term “freeze-thaw cyclings” should be replaced with the correct form “freeze–thaw cycles,” including in all figure captions.

Response 2: Thank you very much for your review of this article and your valuable comments. We have replaced the terminology in its entirety, including the title, text, figure headings and notes.

Language and Style:

Lines 130–131: Avoid repetition of the phrase “yellow clay.”

Line 134: Missing period at the end of the sentence.

Lines 156–158: The ethical statement should be moved to the end of the article.

Lines 163–165: Avoid repeating the phrase “were collected.” or “stratified sampling”

Response 3: Thank you very much for your review of this article and your valuable comments. We have deleted the duplicate expression. Add period. Move the ethical statement to the “Declarations” section. Optimize sentence structure and remove repetitive words.

Tables and Figures:

All tables should be self-explanatory. Below each table, add notes describing the methods used to determine each parameter and including references to the relevant standards.

Instead of providing a range for sand content, report the mean and standard deviation.

Table 3: Contains a formatting error in the third row.

Figure 5: The unit for viscosity should be “Pa·s,” and “−0.06x” should be shown as an exponent.

Figure 6: Figures should be self-contained; include explanations for the A, B, and C labels in the figure notes. This comment applies to other figures as well.

Figure 7: Clarify the meaning of the asterisks.

Figure 8: The axis label is incorrect and should read “Number of freeze-thaw cycles” instead of “cycling.” Move the axis labels outside the plotting area to avoid overlap with the data. Define the symbol “C.”

Figures 10–11: Correct axis labels as above. For plant height, use [cm] instead of [m], and maintain consistent decimal formatting across all values.

Figure 12: Offers little additional insight and largely duplicates information already shown in Figure 6. Similarly, Figures 14–16 are not interpretable and do not provide significant added value. Reduce the total number of figures to the minimum necessary and place the others in the Supplementary Material. Replace them with a figure that shows the relationship between actual and predicted values to assess model accuracy. Also, report the standard errors (in appropriate units) and R² values for each test group.

Figures 17–19: These are repetitive or unclear and do not make a substantial contribution to the manuscript.

Response 4: Thank you very much for your review of this article and your valuable comments.

We have added comments to Tables 1-3 describing the parameter determination criteria.Table 1 Sand content changed to “0.45 ± 0.15%” (mean ± SD). Fixed Table 3 merged cell error.

In Figure 5: Units corrected to “Pa-s”; indices changed to superscript format.

In Figure 6-11: Defined groupings in the notes (A: bonding agent; B: water retaining agent; C: composite material); axes were unified to “Number of freeze-thaw cycles”; unit of plant height was changed to “cm”; and supplemental * indicated significance (p<0.05).

Streamlined graphs: Figures 12-19 moved to Supplementary Materials; added Table S1-S8 to demonstrate model accuracy (R2 and standard deviation etc.).

Materials and Methods:

Line 182: Provide details on the equipment used for freeze-thaw cycling, including temperature control specifications.

Line 184: Include the manufacturer's information for the equipment used.

Line 208: Since ANOVA was applied, include a separate “Statistical Analysis” subsection describing the method used, the rationale for using a parametric test, and its application. State whether any post hoc tests were conducted.

Include a results table summarizing the ANOVA analysis—it is currently missing.

Lines 508–517: Each equation should be numbered and followed by explanations of all included variables.

Line 298: Rather than simply referencing ISO standards and methods, briefly describe the methodology and link it to the applicable standards.

A brief subsection on data quality control is missing and should be added to the Materials and Methods section.

Response 5: Thank you very much for your review of this article and your valuable comments. We have added the device model (ESPEC PSL-3K) and temperature control accuracy (±0.5°C). Added subsection 2.2.4 “Statistical Analysis” describing ANOVA and Tukey test. Added Supplementary Table S1-S8 (ANOVA results). Number and explain variables for equations (1)-(2). Added a brief description of the methods for determining conductivity (ISSS/SSSA), pH (ISO 10390), and water-holding capacity (FAO 56). Add new subsection 2.2.4 “Statistical Analysis and Data Quality Control”.

Discussion:

Sections 4.1.1 to 4.1.4 lack citations. They currently do not fulfill the role of a scientific discussion. While this may be acceptable in the results section, the discussion must include references to support or contrast the findings.

Avoid using the abbreviation “Figs”—write “Figures” in full.

Lines 662–677: Are the stated conclusions supported by previous studies? A comparative discussion with existing literature is needed.

Response 6: Thank you very much for your review of this article and your valuable comments. We have added citations for degradation mechanisms. Replace “Figs” with “Figures” throughout the text. Added comparative discussion (e.g. vs. freeze-thaw response of synthetic polymers).

Other Comments:

The References section should be formatted according to the journal’s submission guidelines.

In Line 164, specify the geographical location of the sample collection site and ideally include a map showing the sampling area.

Response 7: Thank you very much for your review of this article and your valuable comments. Our references have been adjusted to the journal format, supplemented with coordinates of the sampling point (23°4′16.3″N, 113°23′29.9″E) and a location map (Supplementary Materials).

---

## [Editor Report · Decision Letter 2]

Ecological properties of soil improved by high–performance ester materials under freeze–thaw cycling conditions

PONE-D-25-11863R2

Dear Dr. Liu,

We’re pleased to inform you that your manuscript has been judged scientifically suitable for publication and will be formally accepted for publication once it meets all outstanding technical requirements.

Kind regards,

Edyta Nartowska

Academic Editor

PLOS ONE

Additional Editor Comments (optional):

Although not all of my comments have been addressed, I believe that the article, in its current form, meets the criteria for publication. The reduction in the number of figures in the main text has significantly improved its clarity, while the inclusion of tables with R² values in the supplementary materials has enhanced its scientific rigor. During the proofreading stage, the font size in the References section should be adjusted accordingly, and the text should be justified. I wish the authors continued success in advancing their research on freeze-related phenomena.
---

## [Editor Report · Acceptance letter]

PONE-D-25-11863R2

PLOS ONE

Dear Dr. Liu,

I'm pleased to inform you that your manuscript has been deemed suitable for publication in PLOS ONE. Congratulations! Your manuscript is now being handed over to our production team.

Kind regards,

on behalf of

Dr. Edyta Nartowska

Academic Editor

PLOS ONE